# Slab Grave expansion disrupted long co-existence of distinct Bronze Age herders in central Mongolia

Juhyeon Lee [1,2,10], Ursula Brosseder [3,4,10] ✉, Hyoungmin Moon [1], Raphaela Stahl[5], Lena Semerau [5], Jamiyan-Ombo Gantulga[6], Jérôme Magail [7], Jan Bemmann[4], Chimiddorj Yeruul-Erdene[8], Christina Warinner [5,9] ✉ & Choongwon Jeong [1,2] ✉

Dairy pastoralism reached Mongolia during the Early Bronze Age and flourished in the Late Bronze Age alongside the emergence of diverse mortuary practices, including the Deer Stone-Khirgisuur Complex and figure-shaped/Ulaanzuukh burials. While the spread of pastoralism has been widely studied, interactions between these pastoralist groups with distinct mortuary traditions remain underexplored due to challenges in obtaining both genomic and mortuary data. In this study, we analyzed genome-wide and mortuary data from 30 ancient individuals in central Mongolia, a key region where pastoralists with distinct mortuary practices converged. We identify two genetically distinct clusters persisting throughout the Late Bronze Age that correspond to separate burial types, suggesting limited genetic mixing and a maintenance of distinct mortuary practices despite their coexistence. These groups were eventually replaced during the Early Iron Age by the expansion of the Slab Grave population and the establishment of a new burial tradition. Finally, we refine the genetic origin of the Late Bronze Age Deer Stone-Khirgisuur Complex populations, tracing their minor western Eurasian ancestry back to the Eneolithic/Early Bronze Age Afanasievo and Early Bronze Age Khemtseg (Chemurchek) populations. This study provides fine-scaled genetic tracking of major mortuary transitions in prehistoric Mongolia, offering insights into the complex and divergent processes that shaped the ancient pastoralist societies of Asia.

The eastern Eurasian Steppe, encompassing present-day Mongolia and surrounding regions, has served as a nexus of population movement and cultural interaction since prehistoric times. Reflecting this dynamic history, recent studies of ancient genomes in the eastern Eurasian Steppe have revealed distinct genetic profiles associated with various archaeological traditions[1,2]. For example, during the Eneolithic (ca. 3000-2600 BCE) and the Early Bronze Age (EBA; ca. 2600-2000 BCE), a population originating in the North Caucasus region of the western Eurasian Steppe migrated to Mongolia and introduced mobile dairy pastoralism[1,3–5], a highly productive subsistence strategy in

[1]School of Biological Sciences, Seoul National University, Seoul, Republic of Korea. [2]Institute for Data Innovation in Science, Seoul National University, Seoul, Republic of Korea. [3]Leibniz-Zentrum für Archäologie, Prehistory, Mainz, Germany. [4]Department of Pre- and Early Historical Archaeology, Rheinische Friedrich-Wilhelms-Universität, Bonn, Germany. [5]Max Planck Institute for Evolutionary Anthropology, Leipzig, Germany. [6]Institute of Nomadic Archaeology, Department of Anthropology and Archaeology, National University of Mongolia, Ulaanbaatar, Mongolia. [7]Musée d'Anthropologie préhistorique de Monaco, Monaco, Monaco. [8]Cultural Research Analysts Inc, Henrico, VA, USA. [9]Department of Anthropology, Harvard University, Cambridge, MA, USA. [10]These authors contributed equally: Juhyeon Lee, Ursula Brosseder. ✉e-mail: ursula.brosseder@leiza.de; christina_warinner@eva.mpg.de; cwjeong@snu.ac.kr

steppe environments. These people, associated with the Afanasievo culture (ca. 3300-2700 BCE), are represented in Mongolia by only a few burials[6,7]. However, their genetic legacy continued into the subsequent Khemtseg (Chemurchek) populations (ca. 2600-2000 BCE; hereafter "Khemtseg"), which are known from sites in the Dzungarian Basin and the western Mongolian Altai[1,8–12]. The Middle Bronze Age (MBA; ca. 1800-1500 BCE) remains less well-documented and is represented by a limited number of Mönkhkhairkhan burials found in western and northern Mongolia[13]. By the Late Bronze Age (LBA; ca. 1500-1000 BCE), pastoralism had become the dominant form of subsistence in Mongolia, with burials from this period discovered throughout Mongolia. Archaeological studies have identified two distinct mortuary traditions in LBA Mongolia, each with largely non-overlapping geographical distributions: figure-shaped and related graves in eastern and southern Mongolia, and mounded burials associated with the Deer Stone-Khirgisuur Complex (DSKC) culture in western, northern, and central Mongolia (Fig. 1 and Supplementary Fig. 1).

Figure-shaped graves (ca. 1450-1000 BCE) are predominantly found in eastern and southern Mongolia, along with Ulaanzuukh and dumbbell-shaped graves, variants which appear only at a limited number of sites (Fig. 1 and Supplementary Fig. 2)[14,15]. While a variety of grave forms have been documented, they all share archaeological features indicating that they represent related traditions (Supplementary Fig. 2 and Supplementary Note 1)[14,15]. Genetic data from individuals in Ulaanzuukh burials have shown that their ancestry is predominantly derived from Ancient Northeast Asians (ANA), a genetic group that was widely distributed among hunter-gatherers in northeast Asia during prehistoric times[1]. To date, no ancient genomes have been reported for figure-shaped burials, other than Ulaanzuukh.

Mounded burials, especially Sagsai burials and Khirgisuurs, as well as monuments such as deer stones, are commonly found in western, northern and central Mongolia (Fig. 1) and they are attributed to the DSKC culture (Supplementary Fig. 2 and Supplementary Note 1)[14,16–18]. Previous genetic studies have divided individuals associated with the DSKC culture into two groups based on distinct genetic profiles: "Khovsgol_LBA" from northern Mongolia and "Altai_MLBA" from western Mongolia[1,3]. Khovsgol_LBA derives most of its ancestry from a local gene pool in the Lake Baikal region, as reported in the Late Neolithic and Bronze Age Serovo-Glazkovo culture, together with a smaller amount of western Eurasian ancestry. Meanwhile, Altai_MLBA individuals have a mixed ancestry profile of Khovsgol_LBA and a western Eurasian gene pool associated with MLBA Andronovo groups[1,3].

These Bronze Age mounded burials and their unique genetic profiles faded with the emergence of the Slab Grave culture, as the Early Iron Age (EIA; ca. 1000-300 BCE) began. The Slab Grave culture (ca. 1100-400 BCE) emerged from the figure-shaped graves in eastern and southern Mongolia and expanded to central and northern Mongolia as far north as the Lake Baikal region, replacing the former DSKC culture. Interestingly, stone materials from Khirgisuurs and some deer stones were repurposed to build Slab graves[15,19,20]. Ancient genome data from northern Mongolia and the Lake Baikal region has shown that the appearance of the Slab Grave genetic profile is associated with the demise of Khovsgol_LBA populations during the EIA[3], mirroring the mortuary turnover observed in the archaeological record.

While archaeogenetic studies have provided valuable insights into the broader genetic landscape of Mongolia during LBA and EIA[1,3], the complex local interactions among various genetic and mortuary groups remain largely unexplored. For example, the reportedly strong genetic separation between the distinct LBA pastoralist groups (i.e., DSKC vs. figure-shaped) may be exaggerated due to the limited and geographically dispersed sampling of ancient genomes. Their genetic interactions require more detailed investigation, particularly within regions where these groups coexisted. Additionally, the rise and spread of the Slab Grave culture has yet to be sufficiently examined in its initial contact zone with the preceding LBA mounded burials.

Central Mongolia, located at the crossroads of two different LBA burial styles and the front wave of the Slab Grave expansion, provides a superb opportunity to investigate prehistoric population dynamics and its correlation in mortuary practices in a key interaction region. It can be viewed as an arena where pastoralist groups—distinct in their genetic profiles and mortuary traditions yet economically similar— came together to interact within a shared ritual landscape, allowing exploration of the dynamics of mortuary and population change during a "clash of civilizations"[21].

In this study, we report 30 new ancient genomes from central Mongolia. Among these individuals, 27 individuals are associated with LBA and EIA burials, including LBA figure-shaped and related graves ($n = 3$), LBA mounded burials (DSKC) ($n = 16$), and EIA Slab graves ($n = 8$), enabling the genomic analysis of the complex mortuary dynamics during these periods. Three other individuals date to succeeding periods and are associated with the Xiongnu ($n = 1$), Uyghur ($n = 1$), and Mongol ($n = 1$) empires, respectively. While modest in size relative to the broad temporal and geographic scope, the dataset offers valuable insights into mortuary and genetic transitions in a key interaction zone of Mongolia. We show that individuals associated with figure-shaped and DSKC mounded burials rarely mixed genetically, despite their spatial and temporal coexistence in central Mongolia for almost 500 years during the LBA. We also reveal that the mortuary transition in central Mongolia, from LBA figure-shaped and DSKC to EIA Slab Grave, was also associated with a genetic turnover to Slab Grave-related genetic profiles involving minimal genetic interaction with prior DSKC groups. Furthermore, we reveal a genetic connection between the LBA DSKC population and the Eneolithic/EBA Afanasievo and EBA Khemtseg populations, highlighting a lingering genetic legacy of the first pastoralists for over two millennia in the eastern Eurasian Steppe.

## Results

### Ancient genome-wide data from the Upper Orkhon and North Tamir River Valleys

We generated genome-wide data for 30 individuals excavated from six archeological sites in two river valleys in central Mongolia: Ar Bulan (ABL; $n = 1$), Ar Modny Adag (AMY; $n = 1$), Maikhan Tolgoi (MKT; $n = 15$), and OOR-284 (SOV; $n = 1$) from the Upper Orkhon Valley, and Khuruugiin uzuur (KHG; $n = 7$) and Tsats Tolgoi (TST; $n = 5$) from the North Tamir Valley (Fig. 1, Table 1, and Supplementary Data 1). Two additional individuals were tested, but did not yield sufficient human DNA (< 0.09%) for downstream analyses. For the 30 individuals producing successful DNA libraries, we enriched these libraries for a panel ("1240K") of 1,233,013 ancestry-informative single-nucleotide polymorphisms (SNPs) using an in-solution DNA capture method. For 10 individuals with ≥10% endogenous DNA preservation, we also generated whole-genome shotgun sequencing data (Supplementary Data 1). All individuals exhibited a typical deamination pattern of ancient DNA (Supplementary Fig. 3) and had low levels of modern human DNA contamination, with mitochondrial (MT) contamination rates below 5% for all individuals and X chromosome contamination rates below 3% for all 17 males (Supplementary Data 1). Among the 1240K SNP positions, 13,289−1,139,288 SNPs were covered for each individual by at least one high quality read (Supplementary Data 1). For population genetic analyses, we merged the newly generated genotype data in this study with previously published genotype data from ancient[1–3,11,12,22–64] and present-day individuals[33,46,58,65–68] (Supplementary Data 2).

Among the 30 newly genotyped individuals, 16 individuals were excavated from mounded graves associated with the DSKC culture, the predominant archaeological tradition in the region during the LBA. One of these individuals (KHG006) was found in a burial that also exhibits features characteristic of earlier MBA Mönkhkhairkhan graves, and is dated to before 1500 BCE, which is earlier than typical for LBA DSKC graves[14]. Currently, we lack sufficient archaeological and genetic

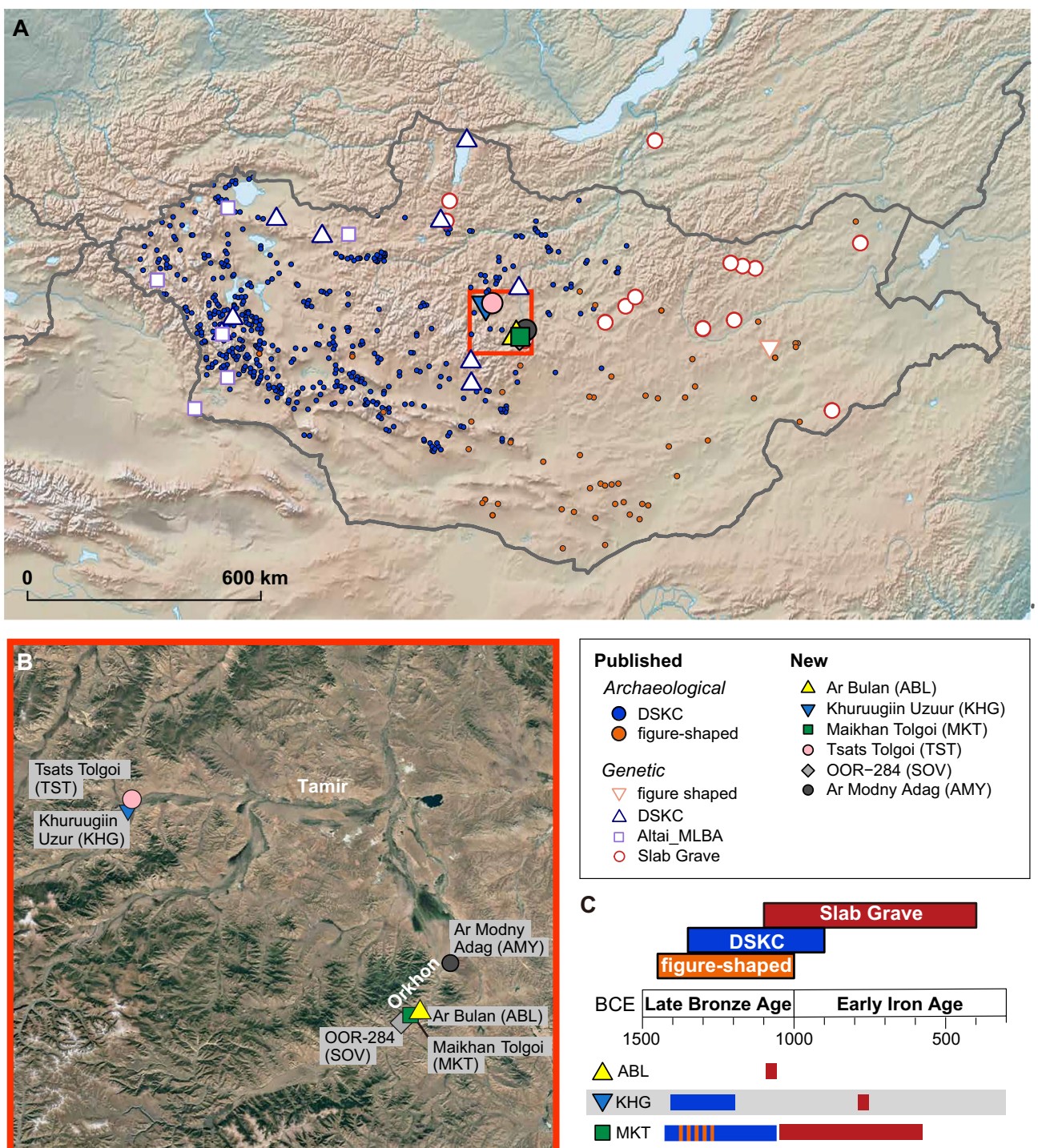

**Fig. 1 | Geographical distribution of LBA/EIA populations in Mongolia.**
**A** Locations of LBA/EIA burial sites in Mongolia. The geographical locations of the LBA burial sites are presented with smaller circles with different colors for each cultural tradition: blue for DSKC and orange for figure-shaped burials. Larger symbols denote sites with genomic data, with empty symbols for previously published sites and filled symbols for the six sites newly analyzed in this study.
**B** Zoomed-in map showing the six sites with newly generated genomic data. The zoom-in map of the region surrounded by a red box in (**A**) is shown along with the two valleys, Tamir and Orkhon. The colors and symbols for six sites match those in (**A**). **C** Chronology of LBA/EIA burial traditions in Mongolia. The time spans of three

main burial traditions in Mongolia are illustrated using different colors: orange for LBA figure-shaped burials, blue for LBA DSKC burials, and red for EIA Slab Grave burials. Of the six newly analyzed sites, four LBA/EIA sites are shown, with symbols and colors matching those in (**A**) and (**B**). Horizontal bars next to each symbol indicate the time spans during which each site was used, with their colors corresponding to specific burial traditions. The MKT site, represented by a green square, contains both figure-shaped and DSKC burials, as reflected by the overlapping orange and blue colors. Source data are available in Supplementary Data 1 and at https://zenodo.org/records/16743201[103].

**Table 1 | Summary of 30 individuals newly sequenced in this study**

| ID | Archaeological ID | Sex | Culture | Genetic analysis group | Cov | Y hap | Date |
|---|---|---|---|---|---|---|---|
| ABL002 | Ar Bulan, gr. 10 | M | Slab Grave | CentralMongolia_EIA_SlabGrave | 0.213 | Q1 | 1218–933 |
| AMY001 | Ar Modny Adag, gr. 1 | F | Mongol | Mongol | 6.789 | – | – |
| KHG001 | Khuruugiin uzuur, gr. 207 | M | Slab Grave | CentralMongolia_EIA_SlabGrave | 0.256 | N1c1 | 810–774 |
| KHG002 | Khuruugiin uzuur, gr. 128 | F | DSKC | CentralMongolia_LBA_DSKC | 0.033 | – | 1375–1059 |
| KHG003 | Khuruugiin uzuur, gr. 127 | F | DSKC | CentralMongolia_LBA_DSKC | 0.234 | – | 1381–1124 |
| KHG004 | Khuruugiin uzuur, gr. 31 | F | DSKC | CentralMongolia_LBA_DSKC | 0.235 | – | 1369–1019 |
| KHG005 | Khuruugiin uzuur, gr. 35 | M | Xiongnu | Xiongnu | 4.113 | C2b1b1 | – |
| KHG006 | Khuruugiin uzuur, gr. 37 | M | DSKC | CentralMongolia_LBA_DSKC | 0.019 | A | – |
| KHG007 | Khuruugiin uzuur, gr. 126 | F | DSKC | CentralMongolia_LBA_DSKC | 0.043 | – | 1500–1314 |
| MKT001 | Maikhan Tolgoi, gr. 103 | F | DSKC | CentralMongolia_LBA_DSKC | 7.329 | – | 1192–924 |
| MKT002 | Maikhan Tolgoi, gr. 32 | M | DSKC | CentralMongolia_LBA_DSKC | 1.456 | Q1a2a | 1403–1129 |
| MKT003 | Maikhan Tolgoi, gr. 23 | F | DSKC | CentralMongolia_LBA_DSKC | 1.204 | – | 1257–1016 |
| MKT004 | Maikhan Tolgoi, gr. 3 | M | Figure-shaped | CentralMongolia_LBA_FigureBurial | 6.298 | N1c | |
| MKT005 | Maikhan Tolgoi, gr. 70 | M | DSKC | CentralMongolia_LBA_DSKC | 2.738 | Q1a2a1 | 1418–1213 |
| MKT006 | Maikhan Tolgoi, gr. 24 | M | Slab Grave | CentralMongolia_EIA_SlabGrave | 2.858 | Q1a1a1 | 1111–911 |
| MKT007 | Maikhan Tolgoi, gr. 67 | M | DSKC | CentralMongolia_LBA_DSKC | 4.997 | Q1a2a1c | 1207–933 |
| MKT008 | Maikhan Tolgoi, gr. 9 | M | DSKC | CentralMongolia_LBA_DSKC | 9.035 | Q1a2a1 | 1262–1021 |
| MKT010 | Maikhan Tolgoi, gr. 5 | M | DSKC | CentralMongolia_LBA_DSKC | 4.678 | Q1a2a | 1500–1291 |
| MKT011 | Maikhan Tolgoi, gr. 7 | F | Slab Grave | CentralMongolia_EIA_SlabGrave | 0.136 | – | 1007–828 |
| MKT012 | Maikhan Tolgoi, gr. 21 | M | DSKC | CentralMongolia_LBA_DSKC_outlier | 0.386 | Q1a | 1532–1325 |
| MKT013 | Maikhan Tolgoi, gr. 65B | F | Figure-shaped | CentralMongolia_LBA_FigureBurial | 0.165 | – | 1491–1268 |
| MKT014 | Maikhan Tolgoi, gr. 65 A | F | Figure-shaped | CentralMongolia_LBA_FigureBurial | 5.500 | – | 1399–1129 |
| MKT015 | Maikhan Tolgoi, gr. 14 | M | Slab Grave | CentralMongolia_EIA_SlabGrave | 7.729 | Q1a1a1 | 1188–912 |
| MKT016 | Maikhan Tolgoi, gr. 13 | M | Slab Grave | CentralMongolia_EIA_SlabGrave | 4.572 | Q1a1a1 | 751–404 |
| SOV001 | OOR-284 | F | Uyghur | Uyghur | 0.610 | – | 660–866 CE |
| TST001 | Tsats tolgoi, gr. A92 | M | Slab Grave | CentralMongolia_EIA_SlabGrave | 3.672 | Q1a1 | 746–390 |
| TST002 | Tsats tolgoi, gr. A94 | F | Slab Grave | CentralMongolia_EIA_SlabGrave | 0.085 | – | 772–476 |
| TST003 | Tsats tolgoi, gr. A123 | M | DSKC | CentralMongolia_LBA_DSKC | 1.121 | O | 1369–1117 |
| TST004 | Tsats tolgoi, gr. A55 | F | DSKC | CentralMongolia_LBA_DSKC | 0.907 | – | – |
| TST005 | Tsats tolgoi, gr. A122 | M | DSKC | CentralMongolia_LBA_DSKC_outlier | 0.075 | CT | 1400–1226 |

Sex is for genetically determined sex (F, female; M, male). The culture column refers to the broad mortuary affiliation used in this study. Detailed information of each grave is shown in Supplementary Data 1. The genetic analysis group refers to the name of the analysis group used for the genomic analysis. The Cov column refers to the mean depth of coverage across the 1,233,013 SNPs in the 1240K panel. The Y hap column refers to the Y haplogroup assignment results for males. The Date column refers to the calibrated radiocarbon date of each individual (95.4% confidence interval) in cal. BCE. The date for the Uyghur-period individual, SOV001, is in cal. CE.

information about the MBA Mönkhkhairkhan due to a scarcity of remains available for study, so we included this individual within the LBA DSKC culture. Three LBA individuals were excavated from burials exhibiting features typical of eastern and southern Mongolia: one from a rectangular figure-shaped grave (MKT004) and two from nested graves (MKT013 and MKT014). While nested graves have only been documented at the MKT site in central Mongolia to date, their archaeological features align closely with those of figure-shaped graves[14], so we have classified them as part of the LBA figure-shaped burial tradition. These findings establish MKT as the only site among the four newly analyzed LBA/EIA sites that contains both DSKC and figure-shaped burial traditions. Eight other individuals were excavated from Slab graves affiliated with the EIA Slab Grave culture. Finally, one individual each was excavated from Xiongnu (KHG005), Uyghur (SOV001), and Mongol (AMY001) period burials (Fig. 1).

**Minimal genetic interaction between distinct cultural groups**
To visualize the genetic diversity present in central Mongolia during the LBA and EIA periods, we first performed principal component analysis (PCA) on the 27 LBA and EIA individuals newly sequenced in this study together with previously published contemporaneous ancient individuals from Mongolia (Fig. 2 and Supplementary Data 2). As demonstrated in the previous study, published LBA and EIA

individuals are grouped into three distinct genetic clusters that align closely with their archaeological backgrounds, with few outliers (Supplementary Data 3). The first cluster, primarily derived from ANA ancestry, includes 12 individuals from LBA Ulaanzuukh, 13 individuals from EIA Slab graves (hereafter "Ulaanzuukh1" and "SlabGrave1," respectively), and 2 individuals affiliated with the DSKC culture. Additionally, one individual from Ulaanzuukh and three from Slab graves show a high affinity with the Khovsgol_LBA cluster, placing them outside the main ANA cluster (hereafter "Ulaanzuukh2" and "SlabGrave2", respectively)[56]. The second cluster consists of 27 individuals associated with Sagsai and Khirgisuur graves of the DSKC culture, including Khovsgol_LBA individuals ($n = 14$), which serve as the representative core group of the DSKC individuals[1–3]. The third cluster represents a genetic cline for Altai_MLBA individuals ($n = 9$)[1–3].

Most of the newly sequenced individuals from central Mongolia overlap with the first two clusters, leaving only two individuals as outliers (Fig. 2A, Supplementary Fig. 4 and Supplementary Data 3). First, 14 individuals, exclusively affiliated with the LBA DSKC culture, overlap with individuals bearing the Khovsgol_LBA genetic profile. Hence, we grouped them as "CentralMongolia_LBA_DSKC", taking into account both their genetic profile and geographic distribution. Second, 11 individuals overlap with the Ulaanzuukh1 and SlabGrave1 clusters, of which three individuals are from LBA figure-shaped/nested graves, and the

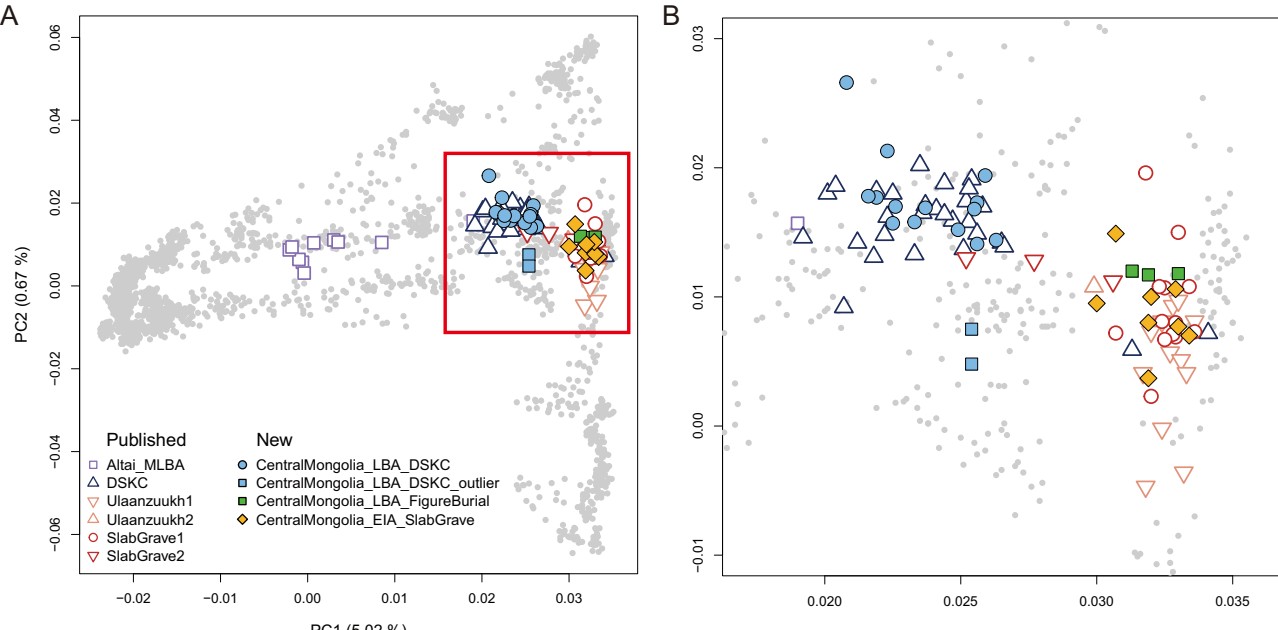

**Fig. 2 | Distinct genetic clusters observed during LBA/EIA periods in Mongolia. A** Principal Component Analysis (PCA) of the 27 newly analyzed individuals from central Mongolia. PCs were calculated using modern Eurasians and ancient individuals from Mongolia and neighboring regions were projected on the calculated PCs. Modern individuals are shown in gray dots. Among the projected ancient individuals, previously published individuals are shown in empty symbols and the newly reported individuals are shown in filled symbols. The colors and shapes of the symbols represent the mortuary tradition and genetic profile of each individual. **B** A zoomed-in view of the red box from (**A**), with *x*-axis ticks ranging from 0.020 to 0.035 (PC1) and *y*-axis ticks from −0.01 to 0.03 (PC2). Point colors and shapes match those in (**A**). Source data are available at https://zenodo.org/records/16743201[103].

remaining eight are from EIA Slab graves. Accordingly, we assigned them into two analysis groups: "CentralMongolia_LBA_FigureBurial" and "CentralMongolia_EIA_SlabGrave". We grouped the remaining two outliers, affiliated with the LBA DSKC culture and slightly displaced downward along PC2 from the other DSKC individuals, as "CentralMongolia_LBA_DSKC_outlier".

It is noteworthy that, apart from two outliers, individuals sharing the same mortuary backgrounds also belong to the same genetic cluster on the PCA. This is evident among the LBA individuals, for whom figure-shaped/nested graves and mounded graves cluster separately. These differences in ancestry and origin may explain the mortuary differences observed in central Mongolia during the LBA, despite both groups occupying the same geographical area and practicing dairy pastoralism. Moreover, a temporal overlap between the two successive genetic clusters, i.e., LBA individuals mostly belonging to the Khovsgol_LBA-like profile while EIA individuals exclusively belonging to the SlabGrave1-like profile, strongly suggests minimal genetic interaction between these groups during a significant population shift in the region. Specifically, Bayesian modeling of AMS dates from LBA Khirgisuurs and EIA Slab graves in the Orkhon Valley indicates that the initial phase of EIA Slab burials could overlap with the final phase of LBA burial traditions for approximately 150 years (Supplementary Figs. 5–7 and Supplementary Note 2)[14].

The clustering pattern observed in PCA was confirmed by the formal test of cladality using the qpWave program[69]. As representatives of the Slab Grave and Khovsgol_LBA profiles outside central Mongolia, we used 13 published Slab Grave individuals without Khovsgol_LBA admixture ("SlabGrave1") and 14 published DSKC individuals from Khovsgol aimag with a homogeneous genetic profile ("Khovsgol_LBA"), respectively. In our qpWave analysis, CentralMongolia_LBA_DSKC is genetically indistinguishable from Khovsgol_LBA, while CentralMongolia_LBA_FigureBurial and CentralMongolia_EIA_SlabGrave are genetically indistinguishable from SlabGrave1 (*p* > 0.381) (Supplementary Data 4). In sharp contrast, CentralMongolia_LBA_DSKC is clearly distinguished from SlabGrave1, and CentralMongolia_LBA_FigureBurial and CentralMongolia_EIA_SlabGrave are distinguished from Khovsgol_LBA (*p* < 1.88 × 10⁻⁴⁸ [48]) (Supplementary Data 4).

Each genetic cluster also had distinct Y-chromosome haplogroup profiles (Supplementary Fig. 8 and Supplementary Data 5). Among the newly sequenced and previously published males associated with the DSKC culture, the majority were assigned to haplogroup Q1a2 (*n* = 17 for Q1a2 and *n* = 2 for Q1a1 out of the total of 28 males), whereas the predominant haplogroup among males linked to the figure-shaped and SlabGrave culture was Q1a1 (*n* = 14 individuals for Q1a1 and *n* = 1 for Q1a2 out of total of 21 males). This contrasting distribution of Y haplogroups further underscores the genetic differences between the two clusters, providing further evidence of a limited genetic interaction during the LBA between people associated with DSKC and figure-shaped burials and a largely complete population replacement by the EIA Slab Grave culture.

Additionally, the CentralMongolia_LBA_DSKC_outlier, the group consisting of two individuals displaced downward along PC2 from the main Khovsgol_LBA-like cluster, shares similar genetic profiles with previously identified outliers: Ulaanzuukh2 (*n* = 1) and SlabGrave2 (*n* = 3)[56]. Like these groups, CentralMongolia_LBA_DSKC_outlier is modeled as the mixture of 24.0% of Ulaanzuukh1 and 76.0% of Khovsgol_LBA ancestry (qpAdm; *p* = 0.073) (Supplementary Data 4)[33], and to our knowledge, it is the first known instance of such a mixed genetic profile among DSKC individuals. One of these outliers (MKT012) was excavated from a platform mound (MKT 21), which features a unique mound design (Supplementary Data 1), mirroring its unique genetic profile[14]. While the identification of the CentralMongolia_LBA_DSKC_outlier reveals that this intermediate genetic profile was present across LBA and EIA mortuary traditions, the number of individuals fitting this profile remains very limited, and no significant shift in the main genetic cluster has been observed, confirming a lack of strong evidence for extensive population mixing.

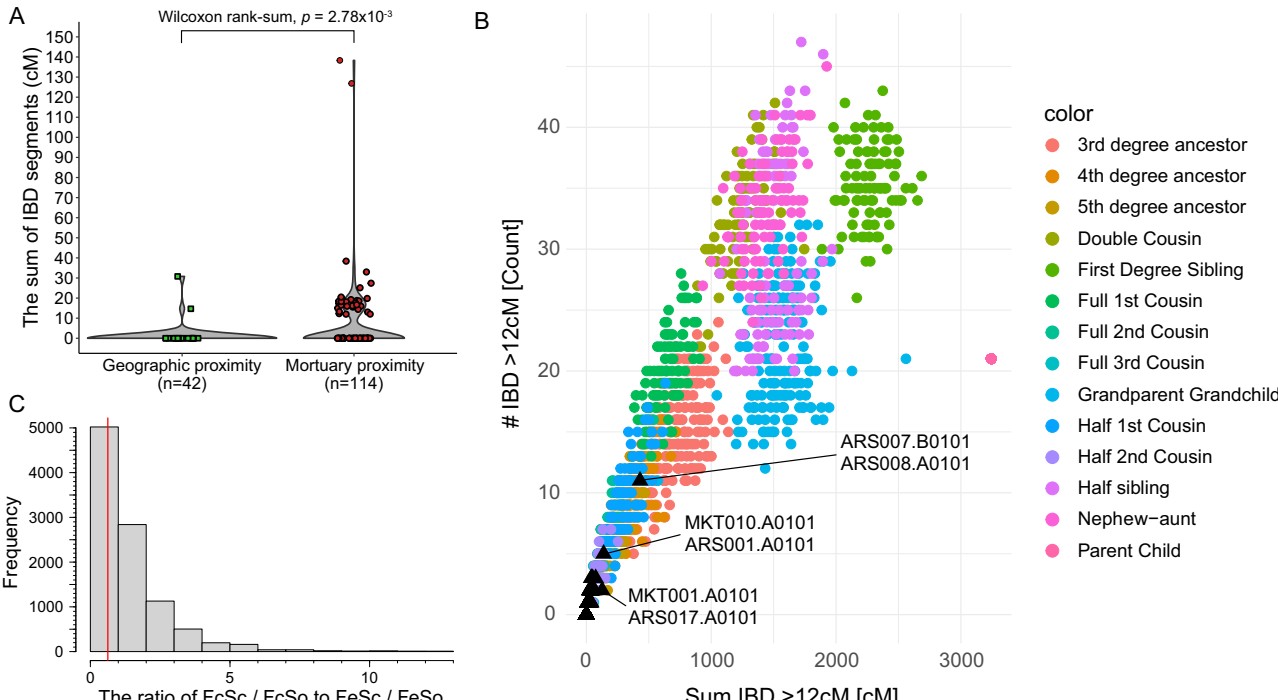

**Fig. 3 | Identity by Descent (IBD) segments shared among individuals from DSKC, figure-shaped and Slab graves. A** The violin plot comparing the sum of IBD segments shared between individuals (1) within the geographic proximity group and (2) within the mortuary proximity group. The geographic proximity group includes pairs from different mortuary traditions (DSKC and figure-shaped) found within 200 km in central Mongolia. The mortuary proximity group includes pairs from the same tradition but from sites >200 km apart, with one individual from central Mongolia. The sum of each pair is shown as colored symbols. The two-sided *p*-value for the Wilcoxon rank-sum test is shown at the top of the plot. **B** Three pairs sharing over 100 cM of IBD plotted against the simulated IBD of genetically related pairs. The *x*-axis represents the sum of IBD segments of at least 12 cM, while the *y*-axis shows the number of these segments. The IBD segments of 14 familial relationships, with 100 replicates simulated using Ped-sim, are represented by distinct colors. Newly analyzed pairs are represented by black triangles, with three labeled pairs sharing over 100 cM of IBD. **C** The distribution of IBD enrichments between

figure-shaped and Slab Grave individuals, generated through permutations. We tested the cultural diffusion model for the Slab Grave expansion in central Mongolia by examining the IBD enrichment shared between figure-shaped individuals from central Mongolia (Fc) and Slab Grave individuals from central Mongolia (Sc). To assess this enrichment, we calculated the ratio of average IBD shared between Fc and Sc (FcSc) to that shared between figure-shaped individuals from eastern Mongolia (Fe) and Sc (FeSc). Each ratio was then normalized using Slab Grave individuals from outside central Mongolia (FcSo and FeSo, respectively) to account for the background IBD level of Sc. A significantly high value of (FcSc/FcSo)/(FeSc/FeSo) would suggest genetic continuity in central Mongolia over time, supporting a cultural diffusion model for the Slab Grave expansion. The ratio was calculated for 10,000 permutations, and the distribution is shown in the figure, with the red vertical line marking the observed value. Source data are available in Supplementary Data 7, 9, and 13, as well as at https://zenodo.org/records/16743201[103].

## Sharing of identity by descent blocks among LBA and EIA herders in Mongolia

The presence of two distinct genetic groups spread out over large geographic areas, one Khovsgol_LBA-like and the other SlabGrave1-like, implies that large-scale population expansions accompanied by a high degree of within-group connectivity had occurred during the LBA and EIA in Mongolia. However, the persistence of these genetic profiles over long periods of time complicates interpretations of population expansion and interaction, as genome-wide allele frequency analyses lack the resolution to detect finer distinctions within these groups. To investigate their genetic interactions and mortuary transitions in more detail, we conducted a higher-resolution analysis by examining Identity by Descent (IBD) block sharing among LBA and EIA individuals in Mongolia using the ancIBD program (Fig. 3 and Supplementary Data 6–8)[70]. Only individuals with sufficient genome-wide coverage were included (*n* = 51 in total; 25 DSKC, 11 figure-shaped, and 15 Slab graves), and only IBD blocks longer than 12 centimorgans (cM) were analyzed in downstream analysis to reduce false positives. We found a total of 169 pairs sharing IBD blocks longer than 12 cM, representing 13.3% of all possible pairs (*n* = 1275), with a cumulative length of 3942.7 cM (Supplementary Data 7 and 8).

Focusing on the LBA in central Mongolia, we observed a strong enrichment of IBD sharing between geographically distant groups with similar mortuary affiliations. Specifically, we analyzed individuals

associated with LBA figure-shaped burials (*n* = 11; 3 figure-shaped and 8 Ulaanzuukh) and LBA DSKC mounded graves (*n* = 25), whose geographical ranges overlapped only in central Mongolia. We compared the patterns of IBD sharing between two groups as follows: (1) pairs consisting of the same mortuary affiliation but from different geographical regions where one of the pair is from central Mongolia (the "mortuary proximity" group), and (2) pairs of individuals from the same geographical region within central Mongolia but with different mortuary affiliations ("geographic proximity" group). Locations within 200 km were considered part of the same geographical area. For both categories, we found that both the total length and the maximum length of the IBD blocks were significantly higher among individuals sharing the same mortuary affiliation (Wilcoxon rank-sum test: $p = 2.78 \times 10^{-3}$ for the sum and $p = 2.54 \times 10^{-3}$ for the maximum) (Fig. 3A, Supplementary Fig. 9 and Supplementary Data 9). When mapping the spatial distribution of shared IBD blocks (Supplementary Fig. 10 and Supplementary Data 10), the mortuary implications of these genetic patterns become even clearer: individuals preserved their mortuary traditions even during long-distance movements and primarily interacted with others who shared similar mortuary practices and genetic profiles, which corroborates the PCA results.

The long-distance connections between individuals with similar mortuary practices are further evidenced by two pairs of distantly located individuals sharing long IBD blocks (MKT001-ARS017 and

MKT010-ARS001, Fig. 3B, Supplementary Figs. 11 and 12 and Supplementary Data 7). Both pairs consist of individuals associated with the DSKC culture, with one individual from the MKT site and the other from the Arbulag sum (ARS) region in northern Mongolia[3]. Despite ~360 km distance between these two sites, each pair shared IBD over 100 cM (Supplementary Fig. 11 and Supplementary Data 7), suggesting they are 4th to 5th degree relatives. This is corroborated by KIN, a tool for detecting genetic relatedness among ancient individuals[71], which shows that MKT001-ARS017 and MKT010-ARS001 are most likely 5th degree relatives (Supplementary Data 11). These two DSKC kinship pairs represent a rare and important finding, with few parallels in ancient DNA literature. Notable comparisons include a 5th degree relative pair from the Afanasievo culture interred 1410 km apart[70], and three Xiongnu-period individuals related at the 3rd–5th degree and buried 350–1000 km apart[72]. These cases have been interpreted as strong evidence for long-distance demic diffusion (Afanasievo) or high mobility among nomadic pastoralists (Xiongnu). The DSKC examples are similarly noteworthy. They provide rare evidence of extended kinship ties across considerable geographic distances and offer compelling support for the high degree of mobility and social cohesion within the DSKC cultural sphere.

Notably, one of the four individuals—ARS017—presents a particularly striking case (Supplementary Fig. 11). Despite being cladal with the figure-shaped and Slab Grave clusters, ARS017 lacks close kinship with them (Supplementary Data 12). Instead, ARS017 shares long IBD blocks with MKT001 from central Mongolia, who is genetically distinct but shares the DSKC mortuary practice. Archaeologically, ARS017 was excavated from a burial definitively associated with the DSKC, located in northern Mongolia—a core region of the DSKC cultural sphere—and dated much earlier than MKT001 (Supplementary Data 1)[3]. Therefore, ARS017 was likely considered a member of the DSKC community. We speculate a scenario in which ARS017 or his close ancestors migrated to and were integrated into the DSKC community, after which inter-marriage with individuals with the DSKC genetic profile over generations diluted the genome-wide signal of admixture in MKT001. ARS017 thus provides a rare insight into mortuary and genetic patterns of interaction during LBA. Overall, however, the patterns of ancestry, kinship, and mortuary affiliation we observe during the LBA suggest that genetic interactions were primarily confined to the same mortuary groups and individuals from different mortuary backgrounds rarely mixed genetically, even at cultural crossroads such as in central Mongolia.

Patterns of IBD sharing also offer insights into the expansion of the EIA Slab Grave culture in central Mongolia. Archaeological studies have proposed that the Slab Grave culture originated from figure-shaped grave groups in eastern Mongolia and rapidly spread into central and northern Mongolia through demic expansion, replacing local LBA populations[15]. If this scenario holds, we expect Slab Grave individuals to share more IBD with each other than with figure-shaped individuals, and vice versa, reflecting limited genetic exchange between groups. To test this, we calculated the average IBD shared within each group and between the two groups, by dividing the total shared IBD length by the number of possible individual pairs. We defined our test statistic as the difference between the mean within-group IBD sharing (averaged across both groups) and the mean between-group IBD sharing, which yielded a value of 3.585 cM. We assessed its significance by constructing a pairwise IBD matrix for 11 figure-shaped and 15 Slab Grave individuals (Supplementary Data 13A and B) and performing 10,000 permutations by randomly shuffling individual labels while preserving the matrix structure. Only 4 permutations exceeded the observed statistic (empirical $p = 4.00 \times 10^{-4}$), indicating the enrichment of IBD sharing within each group and demonstrating the statistical power of our approach.

However, this pattern may be influenced by temporal and geographic separation between the two groups. To minimize these confounding effects, we focused on central Mongolia, where both groups are represented by temporally successive individuals. If population replacement occurred, Slab Grave individuals in this region should show minimal IBD with local figure-shaped individuals. To test this, we divided individuals into four groups: figure-shaped from central Mongolia (Fc; $n = 3$), figure-shaped from eastern Mongolia (Fe; $n = 8$), Slab Grave from central Mongolia (Sc; $n = 6$), and Slab Grave from outside central Mongolia (So; $n = 9$). We then computed the average IBD shared between figure-shaped from central Mongolia and Slab Grave from central Mongolia (FcSc; successive groups from the same region) and compared it with the average IBD shared between figure-shaped from eastern Mongolia and Slab Grave from central Mongolia (FeSc; groups from different regions). To account for background IBD levels in Sc, we normalized the estimates using So. Enrichment of FcSc, the IBD shared between successive groups from the same region, would suggest local genetic continuity, while the absence of such enrichment supports population replacement. The normalized FcSc (3.392/2.901) was smaller than the normalized FeSc (2.394/1.281), and 6895 of 10,000 permutations yielded greater enrichment (empirical $p = 0.690$), suggesting no significant local continuity (Fig. 3C and Supplementary Data 13C and D). This result supports a model of homogeneity among Slab Grave individuals across regions, and reinforce the hypothesis of population replacement in central Mongolia, although further validation with larger sample sizes is needed.

Finally, the eastern origin model predicts elevated IBD sharing between Slab Grave individuals and figure-shaped groups from eastern Mongolia. Contrary to this expectation, we observed greater IBD sharing with figure-shaped individuals from central Mongolia (FcS; 3.097 cM) than from eastern Mongolia (FeS; 1.726 cM), largely driven by one individual, MKT014, who shares IBD segments with several Slab Grave individuals. We assessed this pattern by calculating the FeS/FcS ratio and comparing it to a null distribution generated from all 165 possible permutations (Supplementary Data 13C and E). As expected given the limited sample size, the result was not statistically significant (empirical $p = 0.292$) (Supplementary Fig. 13), highlighting the need for additional data to rigorously test the eastern origin model. Overall, these IBD sharing patterns indicate limited genetic continuity between figure-shaped and Slab Grave populations, consistent with a Slab Grave-driven population replacement scenario, while the eastern origin model remains to be formally tested with larger datasets.

Lastly, we estimated the runs of homozygosity (ROH) blocks for both LBA and EIA individuals[73]. Long ROH blocks indicate recent consanguinity, while an abundance of short IBD blocks suggests a small population size. Most individuals did not exhibit significant lengths of ROH blocks; however, two individuals from graves associated with LBA figure-shaped burials (MKT014 and I12960)[2] and one individual from a Slab grave (I6357)[2] displayed ROH blocks of approximately 20 cM or longer (Supplementary Fig. 14). This pattern, combined with the absence of such signals among DSKC individuals, may suggest that the practice of close-kin marriage was likely confined to the figure-shaped and Slab Grave cultures, or perhaps transmitted from the figure-shaped to Slab Grave culture.

## Genetic contribution of Eneolithic/EBA Afanasievo and EBA Khemtseg to LBA populations

Having found no discernible genetic differences between CentralMongolia_LBA_DSKC ($n = 14$) and the previously published DSKC individuals ($n = 27$) (Supplementary Data 4 and 14), we combined them into a single analysis supergroup "Mongolia_LBA_DSKC" ($n = 41$) to enhance resolution for testing their admixture models using qpAdm. Previous studies with smaller number of individuals modeled them as a mixture of two source groups: Late Neolithic to Bronze Age individuals from Lake Baikal region associated with the Serovo-Glazkovo cultures ("Baikal_LNBA")[37,53,54] and MLBA herders from the western Eurasian

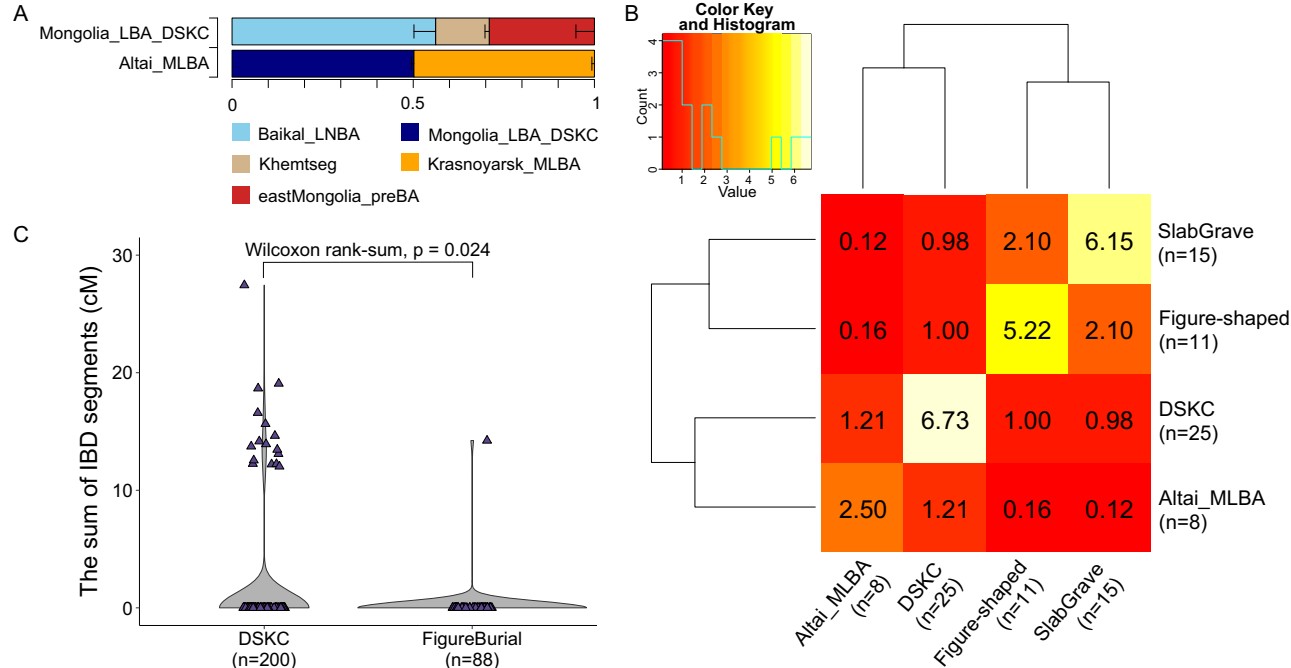

Fig. 4 | The admixture models and IBD sharing of Mongolia_LBA_DSKC and Altai_MLBA individuals. A The admixture models of Mongolia_LBA_DSKC and Altai_MLBA individuals. The colors indicate different ancestry sources used to model Mongolia_LBA_DSKC and Altai_MLBA. The length of the boxes indicates the genetic contribution of the ancestry, and the horizontal bars indicate the one standard errors. B The heatmap of the average length of IBD blocks shared among LBA/EIA individuals. We grouped individuals into four groups, based on their mortuary affiliation and genetic profiles: Altai_MLBA, DSKC (including 23 individuals from Mongolia_LBA_DSKC and 2 culturally DSKC affiliated individuals with

ANA-like genetic profiles), figure-shaped, and Slab Grave. Individuals associated with similar cultures shared more IBD, leading to the clustering of the populations with the similar mortuary affiliation. C The violin plot comparing the total length of IBD shared (1) between Altai_MLBA ($n = 8$) and Mongolia_LBA_DSKC ($n = 25$) and (2) between Altai_MLBA and figure-shaped individuals ($n = 11$). The sum IBD length of each pair is shown as the colored symbols. The two-sided $p$-value for the Wilcoxon rank-sum test is shown at the top of the plot. Source data are available in Supplementary Data 7, 8, 15, and 17.

Steppe (represented by Sintashta_MLBA and/or Krasnoyarsk_MLBA)[48] with the latter contributing <10% ancestry[1,3]. Importantly, for this minor ancestry component, various ancient herder populations distinct from MLBA ones, including earlier EBA populations, fit the model equally well due to limited statistical resolution[1,3].

Testing this model again with our larger sample size and increased resolution using qpAdm, we find that it no longer adequately fits the genetic profile of Mongolia_LBA_DSKC ($p = 1.23 \times 10^{-6}$ for Sintashta_MLBA; $p = 1.58 \times 10^{-6}$ for Krasnoyarsk_MLBA). Moreover, it remains inadequate even after adding an ANA-related population as the third source ($p < 3.62 \times 10^{-3}$; using eastMongolia_preBA, Mongolia_N_North, Ulaanzuukh1, and SlabGrave1 as the ANA proxies)[1,2,56]. This inadequacy becomes even more pronounced when Afanasievo and Khemtseg are added to the outgroup set ($p < 7.51 \times 10^{-4}$ for Afanasievo; $p < 1.80 \times 10^{-4}$ for Khemtseg) (Supplementary Data 15)[1,12,48].

Based on these results, we refined our analysis to obtain two well-fitting three-way admixture models by replacing MLBA herders with local Eneolithic and EBA herder populations, Afanasievo or Khemtseg, and using ANA-related populations as the third source. Mongolia_LBA_DSKC was adequately modeled as a mixture of 66.6 ± 4.9% Baikal_LNBA, 10.5 ± 0.8% Afanasievo, and 22.9 ± 4.4% eastMongolia_preBA ($p = 0.144$) or 56.2 ± 6.0% Baikal_LNBA, 14.8 ± 1.2% Khemtseg, and 29.0 ± 5.1% eastMongolia_preBA ($p = 0.108$). Notably, these three-way admixture models remained robust even when Sintashta_MLBA or Krasnoyarsk_MLBA was added to the outgroup set, supporting no discernible contribution from the MBA Andronovo herders ($p > 0.103$). Also, these models were unanimously applicable to all DSKC subgroups defined by the original study reporting the data set and to all individuals at the individual level (Supplementary Data 16). These findings indicate that the Eneolithic/EBA Afanasievo, the first

herders in the eastern Eurasian Steppe, and the later EBA Khemtseg, who partially descend from the Afanasievo[1,12,48], made important genetic contributions to subsequent LBA DSKC herder populations (Fig. 4A and Supplementary Data 15). We also investigated sex-specific genetic contributions during admixture, prompted by the contrasting Y haplogroup distributions between DSKC and figure-shaped/Slab Grave populations (Supplementary Fig. 8 and Supplementary Data 5). To assess this, we compared admixture proportions between autosomes and the X chromosome (Supplementary Data 15). Unfortunately, we could not obtain reliable ancestry proportion estimates for the X chromosome due to the genetic similarity of the two source groups, Baikal_LNBA and ANA-related populations.

For Bronze Age populations living in far western Mongolia whose ancestry has been described as Altai_MLBA, we recapitulate the plausible proximal model for their ancestry as a mixture of Khovsgol_LBA and Krasnoyarsk_MLBA reported in our previous study[1]. In our modeling, Altai_MLBA ($n = 9$) was well explained as a two-way mixture of 50.0 ± 0.7% of Mongolia_LBA_DSKC and 50.0 ± 0.7% of Krasnoyarsk_MLBA, which serves as a proxy for Western Steppe ancestry ($p = 0.139$) (Fig. 4A and Supplementary Data 15). Comparing results between autosomes and the X chromosome, we found slightly higher DSKC ancestry proportion on the X chromosome, suggesting a slight male-biased contribution from the Western Steppe ancestry (Supplementary Data 15). This pattern aligns with the absence of the Q1a2 Y haplogroup in Altai_MLBA individuals (Supplementary Fig. 8 and Supplementary Data 5).

The sharing pattern of the IBD segments among the MLBA individuals also corroborates a strong genetic connection between Mongolia_LBA_DSKC and Altai_MLBA ($n = 8$; 8 out of 9 individuals have sufficient genome-wide coverage to perform ancIBD), with figure-

shaped individuals more distantly related (Fig. 4B and Supplementary Data 7, 8, and 17). Altai_MLBA shares an average of 1.21 cM of IBD blocks with Mongolia_LBA_DSKC per-pair, which is more than sevenfold higher than the average shared with figure-shaped (0.16 cM) or with Slab Grave (0.12 cM) individuals. In contrast, figure-shaped burial individuals share more IBD blocks with Slab Grave individuals (2.05 cM) than with Altai_MLBA (0.16 cM) and Mongolia_LBA_DSKC (1 cM), further highlighting how cultural connections are reflected in genetic patterns (Fig. 4B and Supplementary Data 17). We also find that both the total length and the maximum length of IBD segments shared between Altai_MLBA and Mongolia_LBA_DSKC exceed those between Altai_MLBA and figure-shaped burial individuals (Wilcoxon rank-sum test: $p = 0.024$ for the sum and $p = 0.024$ for the maximum) (Fig. 4C and Supplementary Fig. 15).

## Discussion

The nature of cultural change–and the degree to which population dynamics versus the diffusion of ideas drives this process–has long been debated within archaeology. Advancing such debates, however, has been challenged by the difficulty of obtaining high-resolution genetic and cultural data from populations experiencing such interactions. Here, we investigated a key instance of cultural change in central Mongolia during LBA and EIA, a region that served as a cultural crossroads for diverse LBA and later EIA groups. During the LBA, culturally differentiated groups with distinct DSKC and figure-shaped burial traditions coexisted for about 500 years before being succeeded by an expansion of the EIA Slab Grave tradition[14].

By integrating paleogenomic and archaeological data, we identified a strong correlation between LBA genetic profiles and mortuary practices. This shows that members of these two distinct mortuary traditions, DSKC and figure-shaped, coexisted but rarely intermingled, maintaining distinct mortuary practices and genetic profiles, despite occupying a shared geographic region and practicing similar subsistence strategies[74]. This genetic distinction is best illustrated at the MKT site in the Upper Orkhon Valley, where burials from both traditions have been found[14]. These two archaeological groups have distinct and internally diverse mortuary practices. DSKC mortuary traditions are associated with mounded burials that include burials of the Sagsai and Khirgisuurs types, while figure-shaped mortuary traditions are associated with figure-shaped, dumbbell, and nested mound graves (Supplementary Fig. 2 and Supplementary Note 1). Overall, we found that individuals primarily interacted with others sharing their broader mortuary tradition, suggesting strong social cohesion despite internal burial variation within each tradition. In addition to burial form, we also found that the orientation of the deceased was closely correlated with genetic profiles, serving as a further marker of social cohesion and likely reflecting different worldviews and belief systems. Specifically, at the MKT site DSKC individuals were consistently buried parallel to the mountain ridge with their heads pointing north-northwest, while all individuals within figure-shaped/nested graves were oriented east-southeast (Supplementary Fig. 2). This pattern of shared genetic ancestry and mortuary traditions within each group seems to be the product of long-standing social norms, including endogamous marriage rules, maintained in the region for several hundred years[14]. This rare LBA archaeological case study of coexistence and the sharing of sacred landscapes with little population mixing over centuries underscores the potential of archaeogenetics to reveal social practices, norms, and choices that are typically elusive in traditional archaeological and genetic analyses.

While our newly generated dataset is limited—particularly for the figure-shaped group, which is expected given its location at the northwestern edge of the distribution—there is currently no evidence to suggest that factors such as gender or social status played a major role in shaping the two distinct genetic groups identified during LBA. In both valleys, we found a roughly equal number of male and female individuals in both the DSKC and figure-shaped groups (Supplementary Data 1). This balanced representation suggests no clear correlation between sex and genetic profile. Indicators of social status are scarce; no grave goods were found, and faunal remains are limited to horse head depositions or burnt sheep bones in khirgisuurs, practices that appear only after 1200 BCE (Supplementary Note 1), making them unlikely to account for earlier differences (1500–1200 BCE). Likewise, limited anthropological data do not indicate notable differences between the groups. Given that the two groups occupy completely different regions, eastern and southern Mongolia for the figure-shaped burials, and western, northern, and central Mongolia for the DSKC, local factors are more likely to have shaped intra-group patterns, rather than explaining the differences between the groups themselves.

With the transition from the LBA to the EIA at the end of the second millennium BCE, DSKC and figure-shaped burials were replaced by the appearance of a new Slab Grave mortuary tradition. This process lasted approximately 150 years (Supplementary Note 2)[14], making it possible to examine in detail the mortuary and genetic dimensions of this turnover. Slab Grave forms include slab, stirrup, and D-shaped burials, with a body orientation to the east (Supplementary Fig. 2 and Supplementary Note 1), similar to that observed in preceding figure-shaped burials. The overlapping distribution of figure-shaped and Slab Grave burials in eastern and southern Mongolia, as well as shared patterns of body orientation, have led to the hypothesis that the Slab Grave phenomenon emerged out of earlier figure-shaped groups. This archaeological hypothesis was corroborated by prior work in eastern Mongolia that genetically linked Slab Grave individuals to earlier Ulaanzuukh burials[1], which are generally classified within the broader figure-shaped burial tradition[14,15]. Here we present genetic data from conventional figure-shaped burials, showing strong evidence that the individuals buried within them have shared genetic profiles with both contemporaneous Ulaanzuukh individuals to the east and succeeding Slab Grave individuals. Further sampling of the figure-shaped and Slab Grave genomes will help understand the emergence and expansion process of the Slab Grave out of the broader LBA figure-shaped meta-population.

Importantly, leveraging on the clear genetic difference between the Slab Grave and DSKC, we show that Slab Grave groups did not genetically intermix with local DSKC groups when they encountered local DSKC groups in central Mongolia. Instead, the DSKC culture and its genetic profile vanished from the region and did not reappear in later periods, suggesting a large-scale displacement. Some DSKC individuals may have migrated northward, contributing archaeologically to emerging EIA populations in Tuva, located in southern Siberia. In Tuva, elite burials such as Arzhan 1 and Tunnug 0 exhibit features rooted in the DSKC culture, including animal-style decoration, monumental grave structures, and the construction of satellites, which were not derived from previous local LBA traditions in Tuva[75,76].

Finally, the data newly generated in our study allowed us to clarify an earlier cultural transition related to population movements during the Eneolithic and EBA. In Mongolia, pastoralism was introduced by populations associated with the Eneolithic/EBA Afanasievo and the EBA Khemtseg cultures[15,77]. Here we find evidence for their continued genetic legacy in the form of a minor genetic contribution to subsequent LBA populations. Specifically, we updated the admixture model of Mongolia_LBA_DKSC as a three-way mixture of Baikal_LNBA, ANA, and Afanasievo/Khemtseg ancestry, pinpointing the western Eurasian source to the Eneolithic/EBA Afanasievo and EBA Khemtseg populations. Notably, since these groups did not have domesticated horses, our refined model suggests that this key technological feature of LBA pastoralism[74,78–80] was likely introduced to the region through cultural transmission. This suggests that throughout most of Mongolia except for the western border on the Altai, LBA pastoralists adopted horse pastoralism without genetic mixing with horse pastoralist groups from the west. This stands in sharp contrast to the adoption of

domesticated horses in the western and central Eurasian Steppe, which was driven by a demic diffusion of people associated with the Sintashta-Andronovo horizon. It remains unresolved why the Sintashta-Andronovo-associated population expansion sharply lost its momentum along the Altai, calling for a focused study on the demographic history and archaeology of the MBA period in Mongolia.

While our findings greatly enhance archaeological understanding of major cultural and genetic transitions in Bronze and Iron Age Mongolia, the transition from the EBA to the MBA remains largely unexplored. Specifically, the emergence of MBA burial traditions, such as Mönkhkhairkhan, the role played by these populations in the spread of horse pastoralism, and their influence on subsequent LBA populations remain unclear due to the limited number of ancient genomes from this period. Despite this scarcity, archaeogenetic studies have identified genetic outliers among MBA individuals, some of which are not found among later LBA populations, which underscore the complexity of dynamics in this era[1,2]. This complexity, compounded by the limited availability of genomes, highlights the urgent need for further targeted archaeological and archaeogenetic investigations to fully elucidate the demographic history of Mongolia.

## Methods

### Sample provenance

Excavations in the Upper Orkhon Valley were carried out between 2009 and 2022 through a collaboration between the Institute of Archaeology, Mongolian Academy of Sciences, and the Institute of Pre- and Early Historical Archaeology, University of Bonn, Germany. The resulting human remains were exported to the Max Planck Institute for Evolutionary Anthropology (MPI-EVA) on 12 March 2019 for scientific analysis under export license A0128802, issued by the Mongolian National Chamber of Commerce and Industry. Excavations in the Upper Tamir Valley were conducted jointly by the Institute of Archaeology, Mongolian Academy of Sciences, and the Musée d'Anthropologie préhistorique de Monaco. The human remains from these excavations were exported to MPI-EVA on 23 October 2018 under export license A0125623, also granted by the Mongolian National Chamber of Commerce and Industry. All procedures for access, sampling, and export were formally approved by the Mongolian National Chamber of Commerce and Industry.

### Archaeological sites and sample description

**Upper Orkhon Valley**. At Maikhan Tolgoi (MKT), the Mongol-German team from Bonn University conducted research from 2009 to 2022. The main cemetery consists of structures located on the southwestern slopes along the mountain ridge Maikhan Tolgoi. In total, 115 funeral structures were documented and over 31 structures were investigated. A series of over 40 radiocarbon dates shows that the cemetery was mainly used in the LBA and EIA. At Ar Bulan (ABL), excavations in 2012 revealed 25 graves located on a ridge along the eastern side of the Orkhon river. Most of the graves belong to the LBA and two Slab graves and two Khirgisuur satellites were excavated. Ar Modny Adag (AMY) is located south of Kharkhorin and 23 km north of Maikhan Tolgoi. The rescue excavation uncovered a burial with a small arrowhead suggesting a Mongol-period date. Additionally, OOR-284 (SOV), 2 km south of Maikhan Tolgoi, was excavated after road damage, revealing a burial from the Late Türkic/Uyghur period. The bone samples from the Upper Orkhon Valley used in the genetic analysis were exported from Mongolia to the MPI-EVA in Germany under the license agreement A0128802.

**North Tamir Valley**. In the North Tamir Valley, the Mongol-Monaco team conducted investigations between 2009 and 2022, with excavations beginning in 2011. Approximately 2100 structures were recorded across a 45,000 hectare area, with most burials dating to the LBA and EIA. At Khuruugiin uzuur (KHG), over 490 structures were

documented, and 12 burials were excavated. At Tsats Tolgoi (TST), 150 structures were recorded, including 6 excavated burials. In Bayantsagaan Valley, 6 burials were uncovered alongside 71 sacrificial structures of Khirgisuur and deer stones. The bone samples from the North Tamir Valley used in the genetic analysis were exported from Mongolia to the MPI-EVA in Germany under the license agreement A0125623.

### DNA library preparation and sequencing

We extracted genomic DNA and prepared single-stranded DNA sequencing libraries of 32 individuals from six sites. During the library preparation, we double indexed the libraries by adding unique 8-mer index sequences at both P5 and P7 Illumina adapters[81]. These DNA extraction and library preparation were conducted in a dedicated clean room facility at MPI-EVA following the previously published protocols[82]. Then we performed shallow shotgun sequencing of these libraries for screening. Among 32 individuals, 30 individuals yielded sufficient amounts of human DNA with >0.09% reads mapped on hs37d5, the human reference genome GRCh37 with decoy sequences. We enriched these libraries for 1,233,013 nuclear SNPs (1240K) by applying in-solution DNA capture techniques with oligonucleotide probes targeting the 1240K sites[28]. These captured libraries were sequenced at MPI-EVA on an Illumina HiSeq 4000 platform to generate 76 bp single-end (SE76) sequences and also at the Bauer Core Facility of Harvard University on an Illumina NovaSeq 6000 platform to generate 100 bp paired-end (PE100) sequences. For 10 individuals with ≥10% human DNA, we prepared whole-genome shotgun sequencing libraries and generated PE100 sequences at the Bauer Core Facility. The output sequences were demultiplexed allowing at most one mismatch in each index. We report the details of the sequencing scheme in Supplementary Data 2.

### DNA sequence data processing and quantification

We removed the Illumina adapter sequences at both ends of the reads and discarded reads shorter than 35 bp using AdapterRemoval v2.3.1[83]. In the same procedure, we merged overlapping read pairs when using PE100 sequencing data. We mapped these reads to hs37d5 using the Burrows-Wheeler Aligner program v0.7.17[84]. We used aln and samse modules. For this, we used the option "-l 9999" to disable seeding and the option "-n 0.01" to allow additional mismatches. To retrieve high-quality reads, we removed polymerase chain reaction duplicates using dedup v0.12.8[85] and then removed reads with the Phred-scaled mapping quality score lower than 30 using SAMtools v1.19.2[86]. We report the summary statistics in Supplementary Data 1.

To authenticate the retained reads, we assessed the postmortem damage pattern and the contamination rate of each library. First, we examined nucleotide misincorporation patterns in the reads using mapDamage v2.2.1[87]. All libraries showed increased C-to-T misincorporation at both ends of the reads, consistent with typical patterns observed in single-stranded libraries (Supplementary Fig. 3 and Supplementary Data 1). Second, we estimated the mitochondrial contamination rate of all individuals using Schmutzi v1.5.7[88] and the X chromosome contamination rate of males using the contamination module of the ANGSD v0.941 (Supplementary Data 1)[89]. All 30 individuals exhibited low levels of modern human DNA contamination, making them suitable for downstream analysis.

Using these processed bam files, we called pseudo-haploid genotypes for 1240K sites by randomly sampling one base at each site and considering the site as homozygous of the chosen base. For this, we used the pileupCaller v1.5.2 program with the "randomHaploid" option (https://github.com/stschiff/sequenceTools). We also used the "singleStrandMode" option to use only positive strand reads to genotype G/A SNPs and negative strand reads to genotype C/T SNPs. We then combined the newly generated genotype data of 30 individuals with the previously published genotype data of modern[33,46,58,65–68] and

ancient individuals[1–3,11,12,22–64], typed on the two sets of SNPs: the Affymetrix Axiom Genome-Wide Human Origins 1 array ("HumanOrigins")[66] and 1240K (Supplementary Data 1).

## Sex determination and uniparental haplogroup assignment

To determine the genetic sex of each individual, we calculated the ratio of sequence coverage between the sex chromosomes and autosomes. For males, the X-to-autosomal coverage ratio is expected to be around 0.5, while for females it is around 1. The Y-to-autosomal coverage ratio is expected to be around 0.5 for males and around 0 for females. Individuals with a Y-to-autosomal ratio >0.3 were classified as male, and those with a ratio <0.1 as female (Supplementary Data 1).

We also retrieved uniparental haplogroups for each individual. For all 30 individuals, we generated mitochondrial consensus sequences with a quality score of ≥10 using the log2fasta program from the Schmutzi package v1.5.7[88], and then we assigned haplogroups of the consensus sequences using HaploGrep v2.1.20[90] (Supplementary Data 1). For the 17 males, we genotyped 13,508 SNPs on the Y chromosome from the ISOGG database using pileupCaller v1.5.2 with the "majorityCall" option. The Y haplogroups were then assigned using a modified version of the yHaplo program[91] (https://github.com/alexhbnr/yhaplo; version 2016.01.08) (Supplementary Data 1).

## Estimation of genetic relatedness

To elucidate genetic kinships among individuals, we calculated the kinship coefficient of every pair of individuals using pairwise mismatch rate (pmr) of pseudo-haploid genotypes. We calculated the pmr between two individuals by dividing the number of sites differently genotyped between the two individuals by the number of sites covered by both[92]. The pmr value with the highest density serves as the pmr for unrelated individuals. We used this value as a baseline to estimate the kinship coefficient. The kinship coefficient for each pair is calculated by one minus the ratio of the pmr of the pair to the baseline. Thus, the pairs with pmr values equivalent to 3/4, 7/8, 15/16 of the baseline categorized as the first, second, and third degree relatives, respectively. We could not find genetic relatives among newly sequenced individuals under the resolution of pmr. To improve the resolution of genetic relatedness estimates for ancient individuals, we also employed KIN v3.1.2, a Hidden Markov Model-based approach that estimates the likelihood of first- to fifth-degree relationships between pairs of individuals[71]. However, we only considered relatedness estimates that were supported by independent IBD analysis, ancIBD[70], to minimize false positives (Supplementary Data 11). We also ran hapROH to identify the ROH blocks, the regions that lack heterozygotes[73]. Long ROH blocks represent a close genetic relationship between the parents of the individual and the short ROH blocks represent the small population size (Supplementary Fig. 14).

## Population genetic analysis

We performed PCA on the combined Human Origins dataset using the smartpca v18140 program from the Eigensoft v7.2.1 package (Fig. 2 and Supplementary Figs. 4 and 16)[93]. By enabling the option "lsqproject: YES", we first computed eigenvectors using 2,077 modern Eurasian individuals and then projected the ancient individuals on the computed eigenvectors (Supplementary Data 2)[33]. The first principal component (PC1) differentiates eastern and western Eurasians, and the second principal component (PC2) differentiates northeast and southeast Asians.

To statistically evaluate the genetic homogeneity of DSKC individuals overlapping on the PCA ($n = 41$), we divided them into three subgroups and conducted a series of $f_4$ symmetry tests. The subgroups were defined as follows: (1) newly reported individuals from central Mongolia (CentralMongolia_LBA_DSKC; $n = 14$), (2) previously reported individuals from Khovsgol Aimag ("DSKC1"; $n = 16$)[3], and (3) geographically dispersed set of DSKC individuals ("DSKC2"; $n = 11$)[2]. The

first two sets were processed in the cleanroom of the Max Planck Institute in Germany while the last was processed in Harvard University. Then, we computed $f_4$-statistics of the form $f_4$(Mbuti.DG, world-wide; X, Y) for all three subgroup-pairs, using 300 world-wide ancient and present-day populations as references. We ran qpDstat v980 in the admixtools v7.0.2 package using the 1240K dataset[66]. Reference populations used in this analysis are listed in Supplementary Data 2, and the full $f_4$ results are provided in Supplementary Data 14.

To additionally confirm the pattern observed in PCA, we performed qpWave and qpAdm analysis (Fig. 4 and Supplementary Data 4, 12, 15, 16, and 18)[33,69]. First, we ran qpWave v1520 in the admixtools v7.0.2 package to test whether two populations overlapping on the PCA are genetically homogeneous. Then we ran qpAdm v1520 in the admixtools v7.0.2 package to model individuals as the mixture of two or three source populations[66]. For both analyses, we used the 1240K dataset and used the following set of populations as the right populations, or outgroups: central African rainforest hunter-gatherer (Mbuti, $n = 5$), Andamanese islander (Onge, $n = 2$), Taiwanese Aborigine (Ami, $n = 2$), Central American (Mixe, $n = 3$), Epipaleolithic Levantine (Natufian, $n = 6$)[33], Neolithic Iranians from the Ganj Dareh archaeological site (Iran_N, $n = 8$)[33,48], Epipaleolithic European hunter-gatherer (Villabruna, $n = 1$)[30], Anatolian Neolithic from the Barcin site (Anatolia_N, $n = 23$)[28]. When multiple combinations of source populations fit the target population, we increase resolution using the "rotating" approach[94]. This involved adding source populations from one model to the right populations when testing alternative source combinations.

To assess potential sex-biased admixture, we performed qpAdm separately for autosomes and the X chromosome. For autosomes, we used the default settings, while for the X chromosome, we specified "chrom:23" in the parameter file. We then quantified sex bias using a Z score, calculated as the difference in ancestry proportions between autosomes and the X chromosome, normalized by their combined standard errors ($z = \frac{P_A - P_X}{\sqrt{\sigma_A^2 + \sigma_X^2}}$, in which $P_A$ and $P_X$ are the admixture proportions on the autosomes and the X chromosome, and $\sigma_A$ and $\sigma_X$ are the corresponding jackknife standard deviations.)[43]. A positive Z score indicates an overrepresentation of a given ancestry in autosomes relative to the X chromosome, suggesting male-biased admixture, whereas a negative Z score suggests a female-biased contribution (Supplementary Data 15).

## IBD sharing analysis

We performed Identity by Descent (IBD) sharing analysis using ancIBD v0.7 to identify IBD blocks shared among LBA/EIA individuals from Mongolia (Supplementary Data 7 and 8)[70]. Before running ancIBD, we first imputed 59 individuals using GLIMPSE (v1.1.1) with the default parameters[95]. Only ancient genomes with sufficient coverage were imputed: 0.25× coverage for shotgun sequenced samples and 1× coverage for 1240K captured samples (Supplementary Data 6)[1–3]. First, for the ancient genomes generated with half-UDG treatment, we trimmed ends of the reads for 3 or 4 base pairs (bp) to remove ancient DNA damage[96], considering the damage pattern obtained by mapDamage[87]. Then we calculated the genotype likelihoods (GL) using bcftools v1.19[97] with the 1000 G panel as a reference[98,99]. For the ancient genomes generated by the single-stranded library preparation, we extracted GLs of C/T SNPs from negative strands, G/A SNPs from positive strands, and the rest from all stands. We then performed imputation using the command "GLIMPSE_phase_static" on genomic chunks of 2,000,000 bp with the buffer size of 200,000 bp. We ligated the chunks using "GLIMPSE_ligate_static" and determined the most likely haplotypes using "GLIMPSE_sample".

Next, we analyzed IBD sharing among imputed individuals using ancIBD v0.7[70]. Specifically, we estimated the pairwise IBD blocks of 59 individuals using the "HapBLOCK" function of ancIBD with default

parameters. Only IBD blocks longer than 8 centimorgan (cM) containing more than 220 SNPs per cM were included in the output. For downstream analysis, we focused on IBD blocks of at least 12 cM to minimize the potential for false positives.

## Kinship estimation based on the IBD sharing

We estimated the kinship of the pairs of individuals using the estimated IBD blocks shared between the pair. For this, we simulated the distribution of IBD segment lengths and counts based on biological kinship and compared the simulated values with the observed values (Fig. 3B). First, we simulated the distribution of IBD segment lengths and counts using Ped-sim[100]. Following the described methods[100], we incorporated a sex-specific recombination map[101] and applied a crossover interference model[102]. Our simulation covered 14 familial relationships, including first-degree (Parent-Child, Siblings), second-degree (Grandparent-Grandchild, Avuncular, Half-Siblings, Double Cousins), first to third-degree full cousins, first to second-degree half cousins, and third to fifth-degree great-grandparent relationships. For each pair of relationships, Ped-sim returns information of the physical and genetic positions of the start and end of each IBD block, as well as the IBD state, determining whether the individuals share one (IBD1) or two IBD haplogroups (IBD2) in the given range.

AncIBD merges the hidden state calls of IBD1 and IBD2 segments[70], while Ped-sim treats adjacent IBD1 and IBD2 segments as separate blocks. We used a custom R script to merge adjacent IBD1 and IBD2 segments in the Ped-sim results. Additionally, we filtered out simulated IBD segments smaller than 12 cM or those containing fewer than 220 SNPs per cM (based on the 1240K SNP panel), which are the filtering threshold used to reduce false IBD segments in our ancIBD analysis.

## Statistical tests for IBD sharing

We performed several statistical tests using R v4.1.2 to compare IBD sharing patterns among LBA and EIA individuals from Mongolia. First, we compared IBD sharing between DSKC and figure-shaped individuals, focusing on two groups: (1) pairs consisting of the one from central Mongolia and the other from a different geographical region, both sharing the same mortuary affiliation (the "mortuary proximity" group), and (2) pairs from the same region within central Mongolia but with different mortuary affiliations ("geographic proximity" group). Locations within 200 km were considered part of the same geographical area. To assess whether differences in the sum and maximum IBD lengths between the two groups were significant, we performed Wilcoxon rank-sum tests using the "wilcoxon.test" function from the R package "stats" (v4.1.2; the package "stats" is part of R) (Fig. 3A, Supplementary Fig. 9, and Supplementary Data 9). We also used the Wilcoxon rank-sum test to compare IBD sharing between Altai_MLBA and other groups (Fig. 4C and Supplementary Fig. 15).

Second, we investigated IBD sharing patterns between figure-shaped and Slab Grave individuals to test the archaeological hypothesis that the Slab Grave culture originated in eastern Mongolia and rapidly expanded westward, replacing earlier populations. To evaluate whether the genetic data support this model, we quantified three IBD enrichment statistics and assessed their significance using permutation-based tests.

Under the replacement hypothesis, we expect higher IBD sharing within each group—figure-shaped and Slab Grave—than between them, reflecting limited gene flow during the expansion. To test this, we constructed a symmetric pairwise IBD matrix for 26 individuals (11 figure-shaped and 15 Slab Grave), using the total length of shared IBD segments in cM (Supplementary Data 13A). For each group, as well as for the between-group comparison, we calculated the average IBD sharing by dividing the total shared length by the number of possible individual pairs. Our test statistic was defined as the difference between the average within-group sharing (averaged across both

groups) and the average between-group sharing. To assess the significance of the observed statistic (3.585 cM) (Supplementary Data 13B), we performed 10,000 permutations by randomly shuffling individual labels while preserving the matrix structure. For each permutation, we recalculated the test statistic and computed an empirical $p$-value as the proportion of permutations with a value equal to or greater than the observed value ($n = 10,000$).

To minimize the confounding effects of temporal and geographic distance on IBD sharing patterns, we focused on central Mongolia, where we have successive genomes from both figure-shaped and Slab Grave individuals. If the Slab Grave culture entered the region through population replacement, we would expect limited IBD sharing between the two groups despite their geographic proximity. In contrast, elevated IBD sharing between them would indicate genetic continuity over time, supporting a cultural diffusion model for the Slab Grave expansion. To test this, we first divided figure-shaped and Slab Grave individuals into four groups, based on their excavation sites: figure-shaped individuals from central Mongolia (Fc; $n = 3$), figure-shaped individuals from eastern Mongolia (Fe; $n = 8$), SlabGrave individuals from central Mongolia (Sc; $n = 6$) and SlabGrave individuals from outside central Mongolia (So; $n = 9$). We then calculated the average IBD sharing between figure-shaped individuals from central Mongolia and Slab Grave individuals from central Mongolia, FcSc, which represents the successive populations from the same region. And we compared it to the average IBD shared between figure-shaped individuals from eastern Mongolia and Slab Grave individuals from central Mongolia, FeSc, which represents the groups from different regions. To account for background IBD levels in Sc, we normalized the estimates using So, resulting in the test statistic of the ratio FcSc/FcSo versus FeSc/FeSo (Supplementary Data 13D). To statistically test whether the observed value is significantly high, we created the matrix showing the total IBD shared between pairs of individuals, with rows representing figure-shaped individuals and columns representing Slab Grave individuals (Supplementary Data 13C), permuted the columns of the IBD matrix 10,000 times, keeping the IBD combination for each individual intact. The empirical $p$-value was then calculated as the proportion of permutations with a ratio equal to or greater than the observed value ($n = 10,000$) (Fig. 3C).

Finally, we tested whether figure-shaped individuals from eastern Mongolia (Fe) contributed more to the Slab Grave population (S) than those from central Mongolia (Fc), as predicted by the eastern origin hypothesis. To do this, we compared the average IBD shared between figure-shaped individuals from eastern Mongolia and Slab Grave individuals (FeS) to that shared between figure-shaped individuals from central Mongolia and Slab Grave individuals (FcS) (Supplementary Data 13E). A significantly high FeS/FcS ratio would support a stronger genetic contribution from eastern figure-shaped groups to the Slab Grave population. To test the significance of the observed value, we permuted the rows of the IBD matrix of the total IBD sharing, with rows representing figure-shaped individuals and columns representing Slab Grave individuals (Supplementary Data 13C). Across all 165 possible permutations, we recalculated FeS/FcS. In one permutation where the denominator was 0, we substituted a value of 0.01 cM to avoid division by zero; this permutation was excluded from the plot due to the extreme resulting ratio (FeS/FcS = 288.8) (Supplementary Fig. 13). The empirical $p$-value was then calculated as the proportion of permutations with a ratio equal to or higher than the observed value ($n = 165$) (Supplementary Fig. 13).

## Reporting summary

Further information on research design is available in the Nature Portfolio Reporting Summary linked to this article.

## Data availability

All data needed to evaluate the conclusions in the paper are present in the paper and/or the Supplementary Materials. All newly generated

sequencing data reported in this study, including raw reads (FASTQ) and aligned reads (BAM), are available from the European Nucleotide Archive under the accession number PRJEB83289. Previously published datasets analyzed in this study are available from the European Nucleotide Archive, with individual accession numbers provided in Supplementary Data 2. The 1240K panel genotype data for the newly generated individuals are available in the Edmond Data Repository of the Max Planck Society at https://edmond.mpg.de/dataset.xhtml?persistentId=doi:10.17617/3.6OUL8B and on Zenodo at https://zenodo.org/records/16743201[103]. The base map in Fig. 1A is in the public domain and accessible through the Natural Earth website (https://www.naturalearthdata.com/downloads/10m-raster-data/). The base map in Fig. 1B is based on Google Earth Pro, version 7.3.6 (image © Maxar Technologies, © 2024 Airbus, © 2024 CNES/Airbus). The base maps in Supplementary Figs. 10 and 12 were created in R v4.3.1 using a publicly available package "maps" v3.4.0. Source data for Figs. 1, 3, and 4, and Supplementary Figs. 9, 10, 13, and 15 are available in the Supplementary Data file. Source data for Figs. 1, 2, and 3, and Supplementary Figs. 3, 4, 5, 6, 7, 8, 11, 12, 14, and 16 are available at https://zenodo.org/records/16743201[103]. The archaeological human remains analyzed in this study are currently housed and curated by the Institute of Archaeology, Mongolian Academy of Sciences, Ulaanbaatar, Mongolia. Access to the remains analyzed in this study can be gained through the Director of the Institute of Archaeology of the Mongolian Academy of Sciences, Dr. Gereldorj Eregzen (https://en.archaeology.ac.mn/c/1032639; info@mas.ac.mn), and in consultation with the original excavators.

## Code availability

All scripts and code used in the analysis are publicly available at https://zenodo.org/records/16743201[103].

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

## Acknowledgements

We thank K. Hölzl and K. Meyer for their assistance with the graphics. This work was supported by the National Research Foundation of Korea (NRF) grant funded by the Korea government (MSIT) (No. RS-2023-00212640 to C.J.), the Global-LAMP program of the National Research Foundation of Korea (NRF) grant funded by the Ministry of Education (No. RS-2023-00301976 to C.J.), the European Research Council under the European Union's Horizon 2020 research and innovation program (804884-DAIRYCULTURES to C.W.), the Werner Siemens Foundation ("Paleobiotechnology" to C.W.), the American School of Prehistoric Research, and the Harvard Radcliffe Institute. The excavations and archaeological analyses in the Upper Orkhon Valley were funded by the German Science Foundation (No. 230723665 (J.B.) and No. 270578681 (U.B.) and the Gerda Henkel Stiftung, AZ11/ZA/12 (Y.CH., U.B.). The excavations in the North Tamir Valley were funded by the Musée d'Anthropologie préhistorique de Monaco (J.M.).

## Author contributions

U.B., C.W., and C.J. designed and conceived the study. U.B, C.W., and C.J. supervised the project. U.B, J.G., J.M., J.B., and Y.CH. provided archaeological materials and resources. R.S. and L.S. generated genetic data. J.L., U.B., and H.M. performed data analysis. J.L., U.B., H.M., C.W., and C.J. wrote the manuscript with inputs from all authors.

## Funding

## Competing interests

The authors declare no competing interests.
