## [Transparent Peer Review file · Nature Communications]

Slab Grave expansion disrupted long co-existence of distinct Bronze Age herders in central Mongolia

Corresponding Author: Dr Choongwon Jeong

Version 0:

Reviewer comments:

Reviewer #1

(Remarks to the Author)

This paper makes a significant contribution to our growing understanding of the genetic and social landscapes of prehistoric eastern Eurasia. The authors present an impressive amount of genetic and archaeological data, and use it to construct a well supported argument for a correlation between archaeological ritual traditions and genetically distinct groups.

While the number of new individuals included is a relatively small sample for a study of population level genetics, it is a considerable number when considering comparable ancient genetics studies. In addition, the use of existing published regional data makes this study more robust from a statistical standpoint, and expands the reach of their conclusions into the preceding Eneolithic and later Iron Age periods.

The evidence presented of genetically distinct populations associated with the DSKC and figure burial traditions is overall compelling. The analyzed individuals sort into DSKC/figure grave groupings across the authors' two primary analyses (PCA and IBD). The focus on individuals buried in different style graves but within overlapping geographic areas (both at the regional/valley and site level in the case of Maikhan Tolgoi), gives strength to the idea that these distinctions are based on social, not geographic, barriers.

The amount of variance explained by the first two principal components is relatively low. This can be an indication of a relatively genetically homogenous population. Despite this, the authors still make a fairly strong case, given the multiple lines of evidence.

The interpretation of what the genetic and archaeological signatures say about social structure and interaction does go beyond what the actual data reflect in some cases. Specifically, the argument that there was "cultural separation" broadly is not supported by the data. There are numerous documented examples of societies where different religious or social groups live side by side and interact in economic, recreational, and political spheres, but maintain strict endogamous marriage practices. The argument that cultural separation existed should be posited as a possibility that can be additionally tested through examination of additional lines of evidence (genetics of faunal assemblages for example could potentially add clarity as to whether economic interaction was also limited), but not as a foregone conclusion.

Two additional social axes that could add additional resolution to the understanding of social dynamics presented here are social status and gender. There is no discussion of whether there is evidence of differential biological affinity for those of different sexes, something which can be reflective of different marriage and matri-/patri-locality patterns. Similarly, there is no discussion of the social status of the individuals analyzed. For example, it is not noted whether individuals were excavated from the primary mound or satellite burials (both of which are present at the Maikhan Tolgoi site). It would strengthen the argument made here if individuals from both primary and satellite burials show similar genetic affinity as between individuals between primary burials. Differential marriage practices are also widely observed between social classes across the archaeological record, and differences in genetic diversity between elite and non-elite classes have been recently documented by many of the current authors in Xiongnu populations.

While there may not be space in this study to address these aspects, these factors should be acknowledged. If there is evidence that there are no genetic distinctions based on these elements, then that would also be a noteworthy conclusion.

There is a significant amount of research on ancient Eurasian genetics being conducted right now. This paper is significant amongst this body of work in its tying of specific archaeological data to genetic landscapes. Much of the current ancient DNA work does not do a good enough job in connecting results to the archaeological record. The authors here have done a good job of bridging these data sets and giving both ancient geneticists and archaeologists usable data and conclusions to test and build upon. I believe that my concerns noted above could be fairly easily addressed by the authors within the current data and analyses, without a major restructuring or rewrite of this paper.

-Elissa Bullion, PhD

Reviewer #2

(Remarks to the Author)

The study by Lee et al titled "Slab Grave expansion disrupted long co-existence of distinct Bronze Age herders in central Mongolia" generates and analyzes genome-wide data for 30 ancient humans from central Mongolia that span the Late Bronze Age to the Early Bronze Age (with three that date to more recent periods). In this study, they performed PCA, IBD, and qpWave/qpAdm analyses, along with cultural comparisons to mortuary tradition, to determine the population composition changes and relationships in central Mongolia. They examined this region because two major archaeological cultures (DSKC and figure-shaped) co-existed during the Late Bronze Age, that seem to have been replaced by a single cultural tradition (Slab Grave). Overall, this study does an in-depth exploration of major genomic trends spanning this time period, that in conjunction with the cultural data, shows complex population dynamics. Below, I include some comments on areas that could have further clarification:

1. Overall, the study does a good job handling many complex archaeological and genetic grouping terms for someone who is not familiar with these groups in the steppe region. However, this can be improved. The most difficult piece was following which of the new sites showed DSKC-, figure-, and Slab Grave-associated individuals. The easiest place to see this is in Supp Fig 4, where it's easy to place the mortuary traditions found at each site, but it doesn't include individuals from all sites (OOR-284, Ar Modny Adag). Its legend described best what I thought could stand out more clearly in the main text, that the MKT site is the only one with both DSKC and figure shaped graves. I think this could be more easily displayed in Figure 1C, where instead of rectangles, or in addition to rectangles, you used symbols like in Supp Fig 4 for individual dates and site, to better show the temporal and geographic overlap between individuals of different cultures at the same site. Additionally, adding the acronyms used throughout the text for the new sites (e.g. MKT) into the figure would help the reader as a reference.
2. For the uniparental analysis (L. 220-227), why was mtDNA variation/grouping not also discussed? Y-chromosome haplogroups separate Q1a1 (SlabGrave-dominant) and Q1a2 (DSKC-dominant). How do these two haplogroups relate more broadly across worldwide populations? Would contrasting mtDNA and Y-chr help say something about male and female movement in these regions?
3. L. 228: Two outliers are noted, and from Supp Fig 4, they come from two different sites but cluster together. Is there anything shared archaeologically by these two individuals? Do they share more ancestry with each other than with other DSKC individuals?
4. The familial relationship to the 4th/5th degree across two sites spanning 360 km seems remarkable. Is there other evidence in the literature of finding this degree of familial connection in other cultures? Also, ARS017 has a different genetic profile – that's even more surprising. Can ARS017 be examined a bit more, to verify they are admixed, presumably a mixture of DSKC- and ANA-related ancestries? Are the shared IBD blocks all largely in DSKC-related genomic regions?
5. The paragraph starting on L. 294 emphasizes use of IBD sharing to study the relationship between Slab Grave and figure-shaped grave groups. Could we verify that this method has the statistical power needed to show a significant result? For example, can you find IBD sharing enrichment when comparing older and younger DSKC groups, or western and central DSKC groups (Supplementary Figure 9 might get at this, but it is not discussed for DSKC in the main text)? Also, the use of the abbreviations (e.g. FcSc) made it hard to read and understand this paragraph and the next. I think this section can be rewritten to make it easier to follow.
6. On L. 317, it is confusing to me why MKT014 shares so many IBD segments with many Slab Grave individuals, and what that means. Is this related to the recent consanguinity noted in the next paragraph? If yes, then wouldn't we expect to see excess IBD segments for I12960 as well? Some greater interpretation of the results here would help.
7. Starting L. 333, a point was made that the DSKC groups are genetically indistinguishable. Is it possible to do an f4-symmetry test to confirm that the supergroup is reasonable? qpAdm can be robust to smaller sample sizes – if the f4-symmetry test does suggest different groups, it might be useful to confirm the qpAdm result would hold for all the subgroups. This way, at least, we can make sure the new qpAdm mixture profile for the supergroup isn't due to some bias from grouping populations that are not the same.

Minor Comments

--L. 79 – first instance of DSKC acronym, perhaps spell it out?

--L. 100 – 'strong separation between LBA pastoralist groups' – is this referring to DSKC vs figure-shaped populations? I

was a bit confused.

--L. 111: report ancient GENOME-WIDE data from 30 individuals? Since not whole genomes?

--L. 113: keep '8 are associated with EIA Slab graves' to same format as previous two 'EIA Slab Graves (n=8)', so easier to understand.

L. 135: Please share coverage range across the dataset

L. 167: Should Figure 2A also be cited for the PCA?

Figure 4B – missing 'p' in figure-shaped label on x-axis. Also, is DSKC here supposed to be 'Mongolia_LBA_DSKC'?

L. 386: first mention of Baitag burial tradition in main text; should that be introduced earlier? It's used very technically here, but context of why the Baitag tradition is important to examine here (and in Discussion?) would be useful.

L. 389: Does the 'archaeological continuity' noted here need a citation?

Reviewer #3

(Remarks to the Author)

The article makes a significant contribution to our understanding of interactions between culturally distinct populations that coexisted or succeeded one another during the Late Bronze Age and Early Iron Age in Mongolia. Through genetic analysis of 30 individuals from archaeological funerary sites (including 19 from the Bronze Age and 8 from the Iron Age), the authors highlight the limited genetic admixture between two populations that coexisted for approximately 500 years within the same territory, yet remained genetically and culturally distinct: the Deer Stone-Khirgisuur Complex (DSKC) population and the figure-shaped burials population.

The analyses provide new insights into the abrupt transition at the beginning of the Iron Age, characterized by the rapid replacement (over approximately 150 years) of earlier populations by those of the Slab Burial culture. This transition, therefore, cannot be explained solely by cultural diffusion. The disappearance of the DSKC culture from the region further suggests a population displacement, possibly northward to present-day Tuva.

The data also help clarify cultural transitions associated with population movements during the Eneolithic and Early Bronze Age by refining models of genetic admixture. Furthermore, the study proposes that the adoption of horseback riding among pastoralist populations resulted from cultural transmission rather than direct migratory influx.

As I am not a geneticist, I cannot assess the validity of the genetic methods used. However, I can emphasize the strength of the research framework, the depth of cultural interpretation, and the robustness of the arguments presented. One of the article's key strengths lies in its excellent archaeological contextualization of the genetic data.

I have few remarks to make.

1. Naturally, there is the issue of the relatively small number of individuals sampled and their representativity across time (long timescales), space (a vast territory), and the complexity of the phenomena at play. However, the authors are well aware of this limitation, which is inherent to the current state of research. Nevertheless, it might be useful to briefly reiterate this constraint in a sentence.

2. Of course, the validity of the archaeological data used can be trusted, but it could be valuable to provide additional archaeological documentation in the supplementary materials, particularly the plans of the unpublished archaeological structures used in this study. If these plans have already been published (excluding grey literature), an alternative solution could be to provide the references in Supplementary Table S1.

3. Although this is not the core focus of the article, I am somewhat surprised by the age estimations provided for the individuals (Supplementary Table S1, column F). More caution is needed regarding the osteological age assessments. It is not possible to determine such precise ages as "20 years" (KTS 31), "25 years" (KTS A92), or "30 years" (KTS E01). Either references should be provided to support such precise estimations, or—preferably—a broader age range should be indicated. Given this, could similar inaccuracies also affect the age estimations of other individuals in the list?

4. Supplementary Figure 2 is clear, but it would be useful to add scales or, at the very least, indicate for readers unfamiliar with Mongolian archaeology that the scales are not the same for the different burial types (DSKC / Slab graves / Figure-shaped burials).

5. Formatting issues:

The text is of good quality, with very few typographical errors or typos. The language appears very clear; however, since English is not my native language, I am not in a position to assess the stylistic quality of the writing.

- Line 404: The word "That" is duplicated in the phrase "This shows that that".

- Line 937: A colon (:) is mistakenly used instead of a slash (/). The correct format should be <https://doi.org/10.1186/s13059-016-0918-z> instead of [//doi.org/10.1186/s13059-016-0918-z](https://doi.org/10.1186/s13059-016-0918-z).

- The DOI formatting in the references should be standardized. DOIs are sometimes provided but are often missing. It would be best to adopt a consistent approach, ideally by including them systematically. This is particularly important because some references in Cyrillic (Refs 10, 11, 12, 13, 16, etc.) are given in transliterated form, which does not facilitate direct bibliographic searches.

Would it be possible to provide the original references instead of (or alongside) the transliteration?

Version 1:

Reviewer comments:

Reviewer #1

(Remarks to the Author)

Overall, the authors responses and edits addressed my previous concerns and comments, and I would like to reiterate that this is a very well written and supported study with significant implications for our understandings of genetic and mortuary landscapes in Mongolia.

There are still places where statements about cultural affiliation would be more accurately framed as mortuary affiliation. For example, in the section on IBD there is the statement, "individuals preserved their cultural traditions even during long-distance movements and primarily interacted with other culturally and genetically similar individuals, corroborating the PCA results." However, there is no evidence presented that other kinds of cultural or economic interactions did not occur, just that genetic interactions were limited and mortuary traditions were conserved/seggregated (as is clearly stated elsewhere by the authors). Another instance is line 512-514 where it states, "Slab Grave groups did not intermix with local DSKC groups". I would recommend explicitly stating that they did not genetically mix. This is an especially important distinction given that later in the discussion, the authors go on to give an example of cultural exchange and interaction without genetic intermixing in their discussion of the diffusion of horse pastoralism.

Otherwise, I do not have any additional comments for the authors.

Reviewer #2

(Remarks to the Author)

The revised manuscript "Slab Grave expansion disrupted long co-existence of distinct Bronze Age herders in central Mongolia" by Lee et al is very well-written and has addressed almost all of my previous questions. This is a well-analyzed and well-written paper that has contributed substantially to the human ancient DNA literature.

The only thing that still bothers me some is the relationship between ARS017 and MKT001 – the reply addressed the ancestry of ARS017, and I just wanted to make sure that MKT001 also has no trace signal of figure-shaped related ancestry? I assume not, by the final conclusion, but I just wanted to check. However, I feel they did sufficient exploration of this shared relationship, and more revision is not needed to address the ancestry quirk here.

Some very minor comments:

L. 166 - 'contains both the DSKC and figure-shaped BURIAL TRADITIONS'  since MKT have nested graves within the figure-shaped burial tradition, I think this language edit will be clearer.

Figure 2 legend – empty and shaded 'marks'  perhaps change to 'symbols' to match language in Figure 1 legend?

L. 213 – This paragraph is very hard to read, I think because it's so technical and uses the group names multiple times. Perhaps revise to make it easier language to follow, e.g. you could say: "During the LBA, the DSKC group is genetically indistinguishable from Khovsgol_LBA, while the figure-shaped group is genetically indistinguishable from SlabGrave1 ($0.4 < p < 0.8$, Supplementary Table 4)." Readers can go to your table for the technical language.

L. 272 – For the phrase: "(1) pairs consisting of the one from central Mongolia and the other from a different geographical region, both sharing the same cultural affiliation (the "cultural proximity" group)", perhaps change it to "(1) pairs consisting of the same cultural affiliation but from different geographical regions where one of the pair is from central Mongolia ("the cultural proximity" group)" for a clearer read?

L. 297-298 – references 67/68 are not incorporated

L. 365 – There is an unfinished sentence – I think the period should be a comma.

L. 416 – Here, but also later in L. 424-425, there should be a 'the' in front of 'X chromosome'

L. 417 – ANA is not a source group but an ancestry right?

L. 531 – no comma should be there after the references

I very much enjoyed reading and reviewing your paper.

--Melinda Yang

Reviewer #3

(Remarks to the Author)

The comments I made during the first review have been addressed, and the authors' responses are satisfactory. I have no further corrections to request.

Point-by-Point Response to Editor's and Reviewers' comments for NCOMMS-24-84887
“Slab Grave expansion disrupted long co-existence of distinct Bronze Age herders in central Mongolia”

We have revised our manuscript to incorporate the reviewers' comments. Please see our detailed point-by-point responses below.

REVIEWER COMMENTS

Reviewer #1 (Remarks to the Author):

Comment 2: This paper makes a significant contribution to our growing understanding of the genetic and social landscapes of prehistoric eastern Eurasia. The authors present an impressive amount of genetic and archaeological data, and use it to construct a well supported argument for a correlation between archaeological ritual traditions and genetically distinct groups.

While the number of new individuals included is a relatively small sample for a study of population level genetics, it is a considerable number when considering comparable ancient genetics studies. In addition, the use of existing published regional data makes this study more robust from a statistical standpoint, and expands the reach of their conclusions into the preceding Eneolithic and later Iron Age periods.

The evidence presented of genetically distinct populations associated with the DSKC and figure burial traditions is overall compelling. The analyzed individuals sort into DSKC/figure grave groupings across the authors' two primary analyses (PCA and IBD). The focus on individuals buried in different style graves but within overlapping geographic areas (both at the regional/valley and site level in the case of Maikhan Tolgoi), gives strength to the idea that these distinctions are based on social, not geographic, barriers.

We appreciate the reviewer's positive assessment of our study. Please see below for our responses to each of your specific comments.

Comment 3: The amount of variance explained by the first two principal components is relatively low. This can be an indication of a relatively genetically homogenous population. Despite this, the authors still make a fairly strong case, given the multiple lines of evidence.

The low amounts of variance explained by the top PCs are inherent in human genetic data due to limited population stratification in humans. For example, Wright's fixation index F_{ST} is only about 0.15-0.2 across world-wide human populations, suggesting that only 15-20% of human genetic variation is allocated to inter-population differentiation (with the remainder allocated to difference between individuals). Therefore, the 5% of total variation in fact represents roughly $\frac{1}{4}$ to $\frac{1}{3}$ of the genetic variation attributable to population differentiation. Therefore, despite the relatively low variance explained by the first few PCs, this level of differentiation is sufficient to identify population structure, supporting our classification of ancient individuals.

Comment 4: The interpretation of what the genetic and archaeological signatures say about social structure and interaction does go beyond what the actual data reflect in some cases. Specifically, the argument that there was "cultural separation" broadly is not supported by the data. There are numerous documented examples of societies where different religious or social groups live side by side and interact in economic, recreational, and political spheres, but maintain strict endogamous marriage practices. The argument that cultural separation existed should be posited as a possibility that can be additionally tested through examination of additional lines of evidence (genetics of faunal assemblages for example could potentially add clarity as to whether economic interaction was also limited), but not as a foregone conclusion.

Thank you for raising this issue. To clarify what we intended to mean, we have changed our wording from "cultural separation" to "mortuary separation", as we cannot infer the nature of daily interactions between these groups. Our key message is that they did not intermarry and likely maintained distinct worldviews or belief systems, as reflected in their burial structures and practices. This study thus presents a rare prehistoric case of strict adherence to endogamous marriage practices between coexisting but distinct in their mortuary tradition: a pattern more commonly documented in historical periods.

Comment 5: Two additional social axes that could add additional resolution to the understanding of social dynamics presented here are social status and gender. There is no discussion of whether there is evidence of differential biological affinity for those of different sexes, something which can be reflective of different marriage and matri-/patri-locality patterns. Similarly, there is no discussion of the social status of the individuals analyzed. For example, it is not noted whether individuals were excavated from the primary mound or satellite burials (both of which are present at the Maikhan Tolgoi site). It would strengthen the argument made here if individuals from both primary and satellite burials show similar genetic affinity as between individuals between primary burials. Differential marriage practices are also widely observed between social classes across the archaeological record, and differences in genetic

diversity between elite and non-elite classes have been recently documented by many of the current authors in Xiongnu populations.

While there may not be space in this study to address these aspects, these factors should be acknowledged. If there is evidence that there are no genetic distinctions based on these elements, then that would also be a noteworthy conclusion.

We appreciate the reviewer's thoughtful comment and agree that gender and social status are important dimensions of social organization. Regarding social status, we would like to clarify that all individuals analyzed in this study were excavated from the central tombs. While Late Bronze Age khirgisuurs (1500–1000 BCE) feature satellite structures, these differ fundamentally from those associated with the later Xiongnu (200 BCE–100 CE); they are not intended for human interment and, when present (1200–900 BCE), contain only animal remains (Supplementary Information L. 53–55). Moreover, there are no artifacts interred with the deceased, and faunal remains are found exclusively in khirgisuur contexts, beginning around 1200 BCE in the form of horse head depositions or burnt sheep bones. The emergence of these faunal deposits in western DSKC khirgisuurs does not account for the strict endogamous marriage patterns observed between the western DSKC and eastern figure-shaped groups between 1500 and 1200 BCE.

In the absence of artifacts or faunal offerings (outside khirgisuurs), the only available proxy for social status is the size or construction effort of a grave - yet this may reflect collective labor or ritual cooperation rather than hierarchical differentiation. Limited anthropological data likewise show no evidence of social stratification. With the data at hand, we see that the monumentalization process that we show in a different article (Bemmann et al., 2024) can be interpreted as indicating emerging differences in social status or rank, but this takes place in the second phase of the LBA within the western group and does not explain the strict marriage rules between the western and eastern group.

With respect to gender, both Late Bronze Age groups and Early Iron Age slab burials show an approximately equal number of male and female individuals, regardless of whether only sites from central Mongolia (Supplementary Table 1) or all sites across Mongolia are considered.

	Central Mongolia		Mongolia in total	
	Male	Female	Male	Female
DSKC	9	7	30	24
Figure-shaped	1	2	7	9
Slab burial	6	2	14	10

This balanced representation does not reveal any clear sex-based differences in genetic ancestry or burial treatment. We have added a sentence in the main text suggesting no clear correlation between sex and genetic profile. Similarly, while the sample is dominated by adult and mature individuals, the presence of two juveniles suggests some age diversity; however, the limited sample size precludes meaningful interpretation of age-related patterns.

Comment 6: There is a significant amount of research on ancient Eurasian genetics being conducted right now. This paper is significant amongst this body of work in its tying of specific archaeological data to genetic landscapes. Much of the current ancient DNA work does not do

a good enough job in connecting results to the archaeological record. The authors here have done a good job of bridging these data sets and giving both ancient geneticists and archaeologists usable data and conclusions to test and build upon. I believe that my concerns noted above could be fairly easily addressed by the authors within the current data and analyses, without a major restructuring or rewrite of this paper.

-Elissa Bullion, PhD

Thank you for your positive assessment of our study.

Reviewer #2 (Remarks to the Author):

Comment 7: The study by Lee et al titled “Slab Grave expansion disrupted long co-existence of distinct Bronze Age herders in central Mongolia” generates and analyzes genome-wide data for 30 ancient humans from central Mongolia that span the Late Bronze Age to the Early Bronze Age (with three that date to more recent periods). In this study, they performed PCA, IBD, and qpWave/qpAdm analyses, along with cultural comparisons to mortuary tradition, to determine the population composition changes and relationships in central Mongolia. They examined this region because two major archaeological cultures (DSKC and figure-shaped) co-existed during the Late Bronze Age, that seem to have been replaced by a single cultural tradition (Slab Grave). Overall, this study does an in-depth exploration of major genomic trends spanning this time period, that in conjunction with the cultural data, shows complex population dynamics. Below, I include some comments on areas that could have further clarification:

Please see below for our responses to each of your specific comments.

Comment 8: 1. Overall, the study does a good job handling many complex archaeological and genetic grouping terms for someone who is not familiar with these groups in the steppe region. However, this can be improved. The most difficult piece was following which of the new sites showed DSKC-, figure-, and Slab Grave-associated individuals. The easiest place to see this is in Supp Fig 4, where it’s easy to place the mortuary traditions found at each site, but it doesn’t include individuals from all sites (OOR-284, Ar Modny Adag). Its legend described best what I thought could stand out more clearly in the main text, that the MKT site is the only one with both DSKC and figure shaped graves. I think this could be more easily displayed in Figure 1C, where instead of rectangles, or in addition to rectangles, you used symbols like in Supp Fig 4 for individual dates and site, to better show the temporal and geographic overlap between individuals of different cultures at the same site. Additionally, adding the acronyms used throughout the text for the new sites (e.g. MKT) into the figure would help the reader as a reference.

Following the reviewer’s suggestions, we have modified Figure 1C to include the temporal information, mortuary patterns, and the site acronyms of each site. We have also added the sentence clarifying that MKT is the only site with both DSKC and figure burials (L. 165-166,

p. 4). We removed OOR-284 and Ar Modny Adag from Supplementary Figure 4, as they do not belong to LBA or EIA.

Comment 9: 2. For the uniparental analysis (L. 220-227), why was mtDNA variation/grouping not also discussed? Y-chromosome haplogroups separate Q1a1 (SlabGrave-dominant) and Q1a2 (DSKC-dominant). How do these two haplogroups relate more broadly across worldwide populations? Would contrasting mtDNA and Y-chr help say something about male and female movement in these regions?

We have added mitochondrial (mtDNA) haplogroup data in Supplementary Table 5B and visualized the composition of Y and MT haplogroups for each group in Supplementary Figure 8. Unlike Y haplogroups, mtDNA haplogroups do not show distinct differences between figure-burial/Slab Grave and DSKC populations.

The most recent common ancestor of all Y chromosomes (Y chromosomal Adam) is estimated to have lived ~250 thousand years ago (kya) (Karmin et al., 2015), while Q1a1 and Q1a2 diverged ~25 kya (Paz Sepulveda et al., 2022). Today, Q1a1 is predominantly found in eastern Eurasia, including Mongolia and Siberia, while Q1a2 is mainly distributed across central Asia, Siberia and the Americas.

To investigate further sex-biased gene flow, we compared qpAdm results from autosomes and X chromosomes for Mongolia_LBA_DSKC and Altai_MLBA. However, due to overall genetic similarity between Baikal_LNBA and ANA sources and limited statistical resolution of X chromosome data, we obtained unacceptably large standard error measures (15-130%) associated with the ancestry proportion estimates. Therefore, we describe the results in great caution in the main text. We have also added results for a similar analysis on Altai_MLBA where more distant sources (Mongolia_LBA_DSKC vs. Sintashta_MLBA/Krasnoyarsk_MLBA) allow accurate ancestry proportion estimates on the X chromosome (Supplementary Table 13).

Comment 10: 3. L. 228: Two outliers are noted, and from Supp Fig 4, they come from two different sites but cluster together. Is there anything shared archaeologically by these two individuals? Do they share more ancestry with each other than with other DSKC individuals?

Archaeologically, the two outliers do not share any features. One, MKT012, is a platform mound burial (Supplementary Information L. 44–48, Supplementary Figure 2), a unique structure with no known parallels in Mongolia to date. The other, TST005, is a typical Sagsai-type burial characteristic of the DSKC group (Supplementary Information L. 32–38, Supplementary Figure 2). These contrasting burial types suggest that, as the dataset grows, we may uncover more complex patterning within the DSKC group. However, at present, there is no clear archaeological basis to interpret the nature of these outliers.

Genetically, the two outliers appear indistinguishable in qpWave analysis (Supplementary Table 4). However, due to the very low sequencing coverage of TST005 (0.075x on the 1240K SNP panel), it is currently impossible to reliably test whether it shares significantly more ancestry with MKT012 than with other DSKC individuals using either IBD or qpWave modeling.

Comment 11: 4. The familial relationship to the 4th/5th degree across two sites spanning 360 km seems remarkable. Is there other evidence in the literature of finding this degree of familial connection in other cultures? Also, ARS017 has a different genetic profile – that’s even more surprising. Can ARS017 be examined a bit more, to verify they are admixed, presumably a mixture of DSKC- and ANA-related ancestries? Are the shared IBD blocks all largely in DSKC-related genomic regions?

We appreciate the reviewer for raising this point. Long-distance kinship between ARS017 and MKT001 is indeed a rare case, with few comparable examples reported: e.g. a 5th degree relative pair from the Afanasievo culture interred 1,410 km apart (Ringbauer et al., 2024) and three Xiongnu-period individuals related to the 3rd to 5th degree and buried 350-1,000 km apart (Gneccchi-Ruscone et al., 2025). Both cases were presented as rare corroborating evidence for either a long-distance demic diffusion of a small population (Afanasievo) or high mobility of nomadic pastoralists (Xiongnu). We have added a description in the main text to highlight the ARS017–MKT001 case.

We performed extra analysis on ARS017 and found no evidence of recent admixture. Using qpWave, we confirmed that ARS017 is cladal with Ulaanzuukh1 and SlabGrave1, while being clearly distinguished from Khovsgol_LBA, recapitulating reports in previous studies (Jeong et al., 2018; Jeong et al., 2020). When modeling ARS017 as a mixture of Ulaanzuukh1/SlabGrave1 and Khovsgol_LBA using qpAdm, we found a negligible DSKC contribution from Khovsgol_LBA ($15.2\pm 7.8\%$ and $12.4\pm 8.4\%$ Khovsgol_LBA contribution when using Ulaanzuukh1 and SlabGrave1 as the first source, respectively) with no significant improvement of the model fit by adding Khovsgol_LBA as the second source (nested $p\text{-value}\geq 0.058$). We provide these results in Supplementary Table S12.

Importantly, ARS017 is not merely a genetic outlier who happens to share long IBD segments with MKT001. Although genetically distinct from typical DSKC individuals, ARS017 was excavated from a burial unambiguously associated with the DSKC, situated within a cemetery composed exclusively of DSKC-type burials in northern Mongolia, a core region of the DSKC cultural sphere, with no evidence of figure-shaped monuments. This archaeological context, together with the shared IBD segments with MKT001, strongly supports that gene flow occurred between distinct mortuary traditions and that ARS017 was likely regarded as a member of the DSKC community. Given that MKT001 is dated significantly later than ARS017, it is plausible that ongoing intermarriage within the DSKC diluted the genome-wide signal of this admixture in MKT001.

Regarding the ancestry background of the IBD segments, we would like to point out that it is currently infeasible to perform accurate local ancestry decomposition between the DSKC and ANA profiles, because they are overall quite similar in the global human genetic diversity context. Still, we tried calculating $f_4(\text{Mbuti.DG}, \text{IBD block}; \text{Khovsgol_LBA}, \text{Ulaanzuukh1/SlabGrave1})$ to test if the IBD blocks shared by ARS017/MKT001 are closer to the DSKC or ANA profiles. Neither of the two IBD blocks showed a significant deviation from zero, indicating no strong genetic affinity to either ancestral profile. We hope that further studies with more DSKC and ANA-associated ancient genomes will revisit this question with sufficient statistical resolution.

Comment 12: 5. The paragraph starting on L. 294 emphasizes use of IBD sharing to study the relationship between Slab Grave and figure-shaped grave groups. Could we verify that this method has the statistical power needed to show a significant result? For example, can you find IBD sharing enrichment when comparing older and younger DSKC groups, or western and central DSKC groups (Supplementary Figure 9 might get at this, but it is not discussed for DSKC in the main text)? Also, the use of the abbreviations (e.g. FcSc) made it hard to read and understand this paragraph and the next. I think this section can be rewritten to make it easier to follow.

To demonstrate that our permutation test has sufficient statistical power, we applied it to IBD sharing patterns between figure-shaped and Slab Grave individuals. While these two groups are genetically indistinguishable in allele frequency-based analyses, they represent successive populations with distinct demographic histories: the figure-shaped individuals precede the Slab Grave individuals, who expanded from eastern Mongolia several hundred years later. Given this temporal separation, we hypothesized that individuals from each group would share more IBD within their group than between groups. To test this, we constructed a pairwise IBD sharing matrix for 11 figure-shaped and 15 Slab Grave individuals. We then computed the average IBD sharing within each group and between the two groups, by dividing the total shared IBD length by the number of possible individual pairs. We defined our test statistic as the difference between the mean within-group IBD sharing (averaged across the two groups) and the mean between-group IBD sharing, which yielded a value of 3.585 cM. To assess the significance of this observed value, we performed 10,000 permutations by randomly shuffling the rows and columns, while keeping individual labels unchanged, and recalculated the test statistic each time. Only 4 permutations produced a value greater than or equal to the observed one, resulting in an empirical p-value of 4.00×10^{-4} . This significant enrichment of IBD sharing within groups supports our hypothesis and confirms that our permutation test has sufficient power to detect subtle but meaningful structure in IBD patterns. We have added these results in the main text.

Comment 13: 6. On L. 317, it is confusing to me why MKT014 shares so many IBD segments with many Slab Grave individuals, and what that means. Is this related to the recent consanguinity noted in the next paragraph? If yes, then wouldn't we expect to see excess IBD segments for I12960 as well? Some greater interpretation of the results here would help.

We think that there is no theoretical link between the ROH signal in MKT014 and the IBD sharing with later Slab Grave individuals. Therefore, we do not expect to see a similar IBD signal in I12960. Overall, the expected distribution of IBD segments between individuals is highly flexible and hard to predict, being affected by many demographic factors including but not limited to population size, marriage practice, and the distribution of individual fitness. While it is indeed intriguing that MKT014 shares many IBD segments, we prefer not to discuss this in detail considering lack of robust theoretical expectation.

Comment 14: 7. Starting L. 333, a point was made that the DSKC groups are genetically indistinguishable. Is it possible to do an f4-symmetry test to confirm that the supergroup is reasonable? qpAdm can be robust to smaller sample sizes – if the f4-symmetry test does suggest different groups, it might be useful to confirm the qpAdm result would hold for all the subgroups. This way, at least, we can make sure the new qpAdm mixture profile for the supergroup isn't due to some bias from grouping populations that are not the same.

We divided the full DSKC set (n=41) into three subgroups: 1) newly reported individuals from central Mongolia (CentralMongolia_LBA_DSKC; n=14), 2) individuals from Khovsgol Aimag reported in Jeong et al. (2018) (“DSKC1”; n=16), and 3) geographically dispersed set of DSKC individuals reported in Wang et al. (2021) (“DSKC2”; n=11). The first two sets were processed in the cleanroom of the Max Planck Institute in Germany while the last was processed in Harvard University.

We computed f4-statistics of the form $f_4(\text{Mbuti.DG, world-wide; X, Y})$ for all three subgroup-pairs, using 300 world-wide ancient and present-day populations as references. Out of 900 total f4-statistics, only 45 showed significant deviations from zero. Among them, 3 were between CentralMongolia_LBA_DSKC and DSKC1, 37 were between CentralMongolia_LBA_DSKC and DSKC2, and 5 were between DSKC1 and DSKC2, respectively. Most populations breaking the symmetry are ancient western Eurasian groups associated with Central Asian/Iranian (e.g. Gonur1_BA), Anatolian (e.g. Anatolia_N), or steppe pastoralist (e.g. Krasnoyarsk_MLBA) ancestries. Consistently, they showed more affinity to DSKC2, implying that DSKC2 may have higher Afanasievo/Khemtseg ancestry than the other two subgroups. Full f4 results are provided in Supplementary Table 14.

Applying the same qpAdm model for the whole DSKC group, we successfully modeled all three subgroups with significant contributions from all three ancestry components. For example, the Baikal_LNBA+Khemtseg+eastMongolia_preBA model yielded $11.0 \pm 1.4\%$, $14.9 \pm 1.3\%$, and $17.3 \pm 1.5\%$ Khemtseg contribution for CentralMongolia_LBA_DSKC, DSKC1, and DSKC2, respectively ($14.8 \pm 1.2\%$ for the whole group). As expected from the f4 results, DSKC2 harbored the highest proportion of the Khemtseg ancestry. Therefore, we firmly believe that the qpAdm model accurately reflects the ancestry composition of the DSKC individuals rather than an artifact of merging heterogeneous groups. Details of the qpAdm results are provided in Supplementary Table 16.

Comment 15: --L. 79 – first instance of DSKC acronym, perhaps spell it out?

The full name is provided in the abstract and in the first paragraph of Introduction (L. 68).

Comment 16: --L. 100 – ‘strong separation between LBA pastoralist groups’ – is this referring to DSKC vs figure-shaped populations? I was a bit confused.

Yes, it is. We have modified the sentence to clarify the meaning.

Comment 17: --L. 111: report ancient GENOME-WIDE data from 30 individuals? Since not whole genomes?

Yes, we generated genome-wide SNP data for 30 individuals using in-solution DNA capture techniques. We generated whole-genome data for 10 individuals with sufficient human DNA through shotgun sequencing.

Comment 18: --L. 113: keep '8 are associated with EIA Slab graves' to same format as previous two 'EIA Slab Graves (n=8)', so easier to understand.

Done.

Comment 19: L. 135: Please share coverage range across the dataset

The coverage range is presented in L. 148.

Comment 20: L. 167: Should Figure 2A also be cited for the PCA?

Done.

Comment 21: Figure 4B – missing 'p' in figure-shaped label on x-axis. Also, is DSKC here supposed to be 'Mongolia_LBA_DSKC'?

We have corrected the typo in Figure 4B. The group labels in Figure 4B are primarily based on the mortuary context because the analysis is intended to investigate the IBD sharing pattern within and between cultural groups. For that purpose, the "DSKC" group in Figure 4B is mostly composed of Mongolia_LBA_DSKC individuals but including two additional individuals (ARS017 and I13505) who are from the DSKC mortuary context but having a figure-burial-like genetic profile.

Comment 22: L. 386: first mention of Baitag burial tradition in main text; should that be introduced earlier? It's used very technically here, but context of why the Baitag tradition is important to examine here (and in Discussion?) would be useful.

Considering the poorly understood archaeological context of the Baitag tradition, we have removed the mentions from the manuscript.

Comment 23: L. 389: Does the 'archaeological continuity' noted here need a citation?

Mentions of the Baitag tradition were removed. Please see our response to comment #22.

Reviewer #3 (Remarks to the Author):

Comment 24: The article makes a significant contribution to our understanding of interactions between culturally distinct populations that coexisted or succeeded one another during the Late

Bronze Age and Early Iron Age in Mongolia. Through genetic analysis of 30 individuals from archaeological funerary sites (including 19 from the Bronze Age and 8 from the Iron Age), the authors highlight the limited genetic admixture between two populations that coexisted for approximately 500 years within the same territory, yet remained genetically and culturally distinct: the Deer Stone-Khirgisuur Complex (DSKC) population and the figure-shaped burials population.

The analyses provide new insights into the abrupt transition at the beginning of the Iron Age, characterized by the rapid replacement (over approximately 150 years) of earlier populations by those of the Slab Burial culture. This transition, therefore, cannot be explained solely by cultural diffusion. The disappearance of the DSKC culture from the region further suggests a population displacement, possibly northward to present-day Tuva.

The data also help clarify cultural transitions associated with population movements during the Eneolithic and Early Bronze Age by refining models of genetic admixture. Furthermore, the study proposes that the adoption of horseback riding among pastoralist populations resulted from cultural transmission rather than direct migratory influx.

As I am not a geneticist, I cannot assess the validity of the genetic methods used. However, I can emphasize the strength of the research framework, the depth of cultural interpretation, and the robustness of the arguments presented. One of the article's key strengths lies in its excellent archaeological contextualization of the genetic data.

I have few remarks to make.

We appreciate the reviewer's positive assessment of our study. Please see below for our responses to each of your specific comments.

Comment 25: 1. Naturally, there is the issue of the relatively small number of individuals sampled and their representativity across time (long timescales), space (a vast territory), and the complexity of the phenomena at play. However, the authors are well aware of this limitation, which is inherent to the current state of research. Nevertheless, it might be useful to briefly reiterate this constraint in a sentence.

We are mindful of the limitations inherent in analyzing a relatively small number of individuals, especially given the long temporal span and the complexity of the cultural and genetic processes under study. To mitigate these constraints, our primary analyses focus on a well-defined micro-region within central Mongolia, which provides a consistent baseline for interpreting patterns before incorporating data from the broader region. We have added a sentence clarifying the limit of relatively small sample size at the last paragraph of Introduction (L. 120).

Comment 26: 2. Of course, the validity of the archaeological data used can be trusted, but it could be valuable to provide additional archaeological documentation in the supplementary materials, particularly the plans of the unpublished archaeological structures used in this study. If these plans have already been published (excluding grey literature), an alternative solution could be to provide the references in Supplementary Table S1.

We appreciate your trust in the archaeological data. We presented the burial plans, typology, and a range of diversity within each mortuary group in Supplementary Figure 2, along with brief details on the burial structures in Supplementary Information 1.

The complete catalogue, accompanied by a detailed analysis, is currently in preparation, and we expect to finalize the manuscript by this fall. Given the scope and significance of this archaeological work, we believe it warrants full publication rather than being relegated to a supplementary section. While I fully understand and value your perspective, it is important to note that publishing the full catalogue as a supplement is not standard practice. Furthermore, the turnaround for genetic analyses is typically faster than for archaeological studies, which tend to be more complex and time-consuming.

Comment 27: 3. Although this is not the core focus of the article, I am somewhat surprised by the age estimations provided for the individuals (Supplementary Table S1, column F). More caution is needed regarding the osteological age assessments. It is not possible to determine such precise ages as "20 years" (KTS 31), "25 years" (KTS A92), or "30 years" (KTS E01). Either references should be provided to support such precise estimations, or—preferably—a broader age range should be indicated. Given this, could similar inaccuracies also affect the age estimations of other individuals in the list?

Thank you for identifying several translation errors in the French reports on the Tamir Valley. Each individual was analyzed in detail by different anthropologists across various field campaigns. For clarity and consistency, we have now standardized all age reporting using a uniform set of age classes, which facilitates a more cohesive comparison across analyses conducted by different researchers.

Comment 28: 4. Supplementary Figure 2 is clear, but it would be useful to add scales or, at the very least, indicate for readers unfamiliar with Mongolian archaeology that the scales are not the same for the different burial types (DSKC / Slab graves / Figure-shaped burials).

Thank you for this input. We changed Supplementary Figure 2 accordingly to add scales for reference.

Comment 29: 5. Formatting issues:

The text is of good quality, with very few typographical errors or typos. The language appears very clear; however, since English is not my native language, I am not in a position to assess the stylistic quality of the writing.

- Line 404: The word "That" is duplicated in the phrase "This shows that that".

Corrected.

Comment 30: - Line 937: A colon (:) is mistakenly used instead of a slash (/). The correct format should be <https://doi.org/10.1186/s13059-016-0918-z> instead of [//doi.org:10.1186/s13059-016-0918-z](https://doi.org:10.1186/s13059-016-0918-z).

We have removed the doi across all references following the Nature formatting guide.

Comment 31: - The DOI formatting in the references should be standardized. DOIs are sometimes provided but are often missing. It would be best to adopt a consistent approach, ideally by including them systematically. This is particularly important because some references in Cyrillic (Refs 10, 11, 12, 13, 16, etc.) are given in transliterated form, which does not facilitate direct bibliographic searches.

Would it be possible to provide the original references instead of (or alongside) the transliteration?

We have added the original references alongside the transliterated forms. However, we are not certain whether these will be retained in the final publication, as this may not align with the journal's standard reference formatting practices (see, for example, <https://www.nature.com/articles/s41586-024-08113-5>).

Point-by-Point Response to Editor's and Reviewers' comments for NCOMMS-24-84887
“Slab Grave expansion disrupted long co-existence of distinct Bronze Age herders in central Mongolia”

We received 15 comments from three reviewers. We present our response to each comment below.

REVIEWER COMMENTS

Reviewer #1 (Remarks to the Author):

Comment 1: Overall, the authors responses and edits addressed my previous concerns and comments, and I would like to reiterate that this is a very well written and supported study with significant implications for our understandings of genetic and mortuary landscapes in Mongolia.

We appreciate the reviewer's positive assessment of our study. Please see below for our responses to each of your specific comments.

Comment 2: There are still places where statements about cultural affiliation would be more accurately framed as mortuary affiliation. For example, in the section on IBD there is the statement, "individuals preserved their cultural traditions even during long-distance movements and primarily interacted with other culturally and genetically similar individuals, corroborating the PCA results." However, there is no evidence presented that other kinds of cultural or economic interactions did not occur, just that genetic interactions were limited and mortuary traditions were conserved/segreated (as is clearly stated elsewhere by the authors). Another instance is line 512-514 where it states, "Slab Grave groups did not intermix with local DSKC groups". I would recommend explicitly stating that they did not genetically mix. This is an especially important distinction given that later in the discussion, the authors go on to give an example of cultural exchange and interaction without genetic intermixing in their discussion of the diffusion of horse pastoralism.

Otherwise, I do not have any additional comments for the authors.

We have changed the sentence to “individuals preserved their mortuary traditions even during long-distance movements and primarily interacted with others who shared similar mortuary practices and genetic profiles, which corroborates the PCA results.”, and “Slab Grave groups did not genetically intermix with local DSKC groups”. Additionally, we have replaced the term “cultural” with “mortuary” where appropriate to improve precision.

Reviewer #2 (Remarks to the Author):

Comment 3: The revised manuscript “Slab Grave expansion disrupted long co-existence of distinct Bronze Age herders in central Mongolia” by Lee et al is very well-written and has addressed almost all of my previous questions. This is a well-analyzed and well-written paper that has contributed substantially to the human ancient DNA literature.

We appreciate the reviewer’s positive assessment of our study. Please see below for our responses to each of your specific comments.

Comment 4: The only thing that still bothers me some is the relationship between ARS017 and MKT001 – the reply addressed the ancestry of ARS017, and I just wanted to make sure that MKT001 also has no trace signal of figure-shaped related ancestry? I assume not, by the final conclusion, but I just wanted to check. However, I feel they did sufficient exploration of this shared relationship, and more revision is not needed to address the ancestry quirk here.

To assess potential traces of figure-shaped-related ancestry, we used a three-way admixture model with Baikal_LNBA, eastMongolia_preBA, and Khemtseg as sources (Figure 4). We compared the proportion of eastMongolia_preBA ancestry—which is genetically indistinguishable from figure-shaped-related ancestry—between MKT001 and six other DSKC individuals from the same site. If MKT001 had received recent gene flow from a figure-shaped population, we would expect a notably higher proportion of eastMongolia_preBA ancestry. However, as shown in the figure below, MKT001 does not exhibit a markedly elevated level of eastMongolia_preBA ancestry, suggesting it did not experience recent gene flow from a figure-shaped population.

In addition, MKT001 shares at least one identity-by-descent (IBD) block longer than 12 cM with four individuals, all of whom are associated with both the DSKC cultural context and genetic profile. genetic profile, except for ARS017.

Some very minor comments:

Comment 5: L. 166 - ‘contains both the DSKC and figure-shaped BURIAL TRADITIONS’ -> since MKT have nested graves within the figure-shaped burial tradition, I think this language edit will be clearer.

Done.

Comment 6: Figure 2 legend – empty and shaded ‘marks’  perhaps change to ‘symbols’ to match language in Figure 1 legend?

Done.

Comment 7: L. 213 – This paragraph is very hard to read, I think because it’s so technical and uses the group names multiple times. Perhaps revise to make it easier language to follow, e.g. you could say: “During the LBA, the DSKC group is genetically indistinguishable from Khovsgol_LBA, while the figure-shaped group is genetically indistinguishable from SlabGrave1 ($0.4 < p < 0.8$, Supplementary Table 4).” Readers can go to your table for the technical language.

We appreciate the reviewer’s suggestion. We have simplified the paragraph to reduce repeated calling of the group names while keeping some details to deliver what we mean clearly to the readers.

Comment 8: L. 272 – For the phrase: “(1) pairs consisting of the one from central Mongolia and the other from a different geographical region, both sharing the same cultural affiliation (the “cultural proximity” group)”, perhaps change it to “(1) pairs consisting of the same cultural affiliation but from different geographical regions where one of the pair is from central Mongolia (“the cultural proximity” group)” for a clearer read?

Done.

Comment 9: L. 297-298 – references 67/68 are not incorporated

Done.

Comment 10: L. 365 – There is an unfinished sentence – I think the period should be a comma.

Done.

Comment 11: L. 416 – Here, but also later in L. 424-425, there should be a ‘the’ in front of ‘X chromosome’

Done.

Comment 12: L. 417 – ANA is not a source group but an ancestry right?

We have replaced the original term “ANA” with “ANA-related populations”.

Comment 13: L. 531 – no comma should be there after the references

Done.

Comment 14: I very much enjoyed reading and reviewing your paper.
--Melinda Yang

Thank you for your positive assessment and valuable feedback.

Reviewer #3 (Remarks to the Author):

Comment 15: The comments I made during the first review have been addressed, and the authors’ responses are satisfactory. I have no further corrections to request.

We appreciate the reviewer’s positive assessment of our study.